# Use of ratiometrically designed nanocarrier targeting CDK4/6 and autophagy pathways for effective pancreatic cancer treatment

Ying Ji[1,2,3,7], Xiangsheng Liu [1,2,7], Juan Li [1,4], Xiaodong Xie[1], Max Huang [1], Jinhong Jiang[2], Yu-Pei Liao[1], Timothy Donahue[5,6] & Huan Meng [1,2✉]

Aberrant cell cycle machinery and loss of the CDKN2A tumor suppressor locus make CDK4/6 a potential target in pancreatic ductal adenocarcinoma (PDAC). However, a vast majority of PDAC cases do not harbor a durable response to monotherapy of CDK4/6 inhibitor. Utilizing remote loading to co-encapsulate CDK4/6 inhibitor palbociclib (PAL) and an autophagy inhibitor hydroxychloroquine (HCQ), we demonstrate a ratiometrically designed mesoporous silica nanoformulation with synergistic efficacy in subcutaneous and orthotopic PDAC mouse models. The synergism is attributed to the effective intratumoral buildup of PAL/HCQ, which otherwise exhibit distinctly different circulatory and biodistribution profile. PAL/HCQ co-delivery nanoparticles lead to the most effective shrinkage of PDAC compared to various controls, including free drug mixture. Immunohistochemistry reveals that PAL/HCQ co-delivery nanoparticles trigger anti-apoptotic pathway after repetitive intravenous administrations in mice. When combined with a Bcl inhibitor, the performance of co-delivery nanoparticles is further improved, leading to a long-lasting anti-PDAC effect in vivo.

[1] Division of NanoMedicine, Department of Medicine, California NanoSystems Institute, University of California, Los Angeles, CA 90095, USA. [2] California NanoSystems Institute, University of California, Los Angeles, CA 90095, USA. [3] Institute of Textiles and Clothing, The Hong Kong Polytechnic University, Hunghom, Kowloon, Hong Kong, China. [4] Key Laboratory of Biomedical Effects of Nanomaterial & Nanosafety, Chinese Academy of Science, 100049 Beijing, China. [5] Department of Surgery, University of California, Los Angeles, CA 90095, USA. [6] Department of Molecular and Medical Pharmacology, David Geffen School of Medicine at UCLA, Los Angeles, CA 90095, USA. [7] These authors contributed equally: Ying Ji, Xiangsheng Liu. ✉email: menghuan@g.ucla.edu

Cyclin-dependent kinase (CDK) 4/6 inhibitors are among a recently developed antitumor therapeutics that target the aberrant cell cycle machinery[1]. CDK4/6 mediates the G1-to-S phase transition by association with cyclin D and further modulating the phosphorylation of retinoblastoma protein (Rb)[2]. Phosphorylated Rb dissociates from E2F transcription factors, freeing them to participate in DNA replication (Supplementary Fig. 1)[2]. The emergence of CDK4/6 inhibitor (CDK4/6i) is offering clinical advances, providing promising therapeutic outcomes in certain breast cancer, such as palbociclib hydrochloride (PAL)[3] in ER+/HER2− breast cancer and ribociclib and abemaciclib in HR+/HER2− breast cancer[4,5]. The success in breast cancer has also yielded great insights into the mechanisms by which tumors evade the cellular homeostasis through the genetic loss or epigenetic silencing of tumor suppressor, which encodes an endogenous inhibitor of CDK4/6 (p16INK4a) to mediate cell cycle progression. Recent studies show that CDK4/6-mediated cell cycle regulation exists in many solid tumor types, including pancreatic ductal adenocarcinoma (PDAC)[6].

While it is reasonable to consider using CDK4/6i beyond breast cancer, it turns out to be challenging because of (i) unsatisfactory efficacy, (ii) drug resistance, (iii) lack of reliable biomarkers for patient selection, and (iv) toxicity[7–9]. In the setting of PDAC, the obstacle also includes the presence of thick dysplastic stroma, preventing intratumoral drug access[10]. It has become popular to consider CDK4/6i combination therapy by introducing a second drug that impacts certain oncogenic pathway, such as mammalian target of rapamycin (mTOR) inhibitors[11,12], autophagy inhibitors[13], mitogen-activated extracellular signal-regulated kinase (MEK) inhibitors[14], immunotherapeutics (anti-programmed cell death ligand 1)[15], transforming growth factor-β (TGF-β) inhibitors[16], and certain chemo-reagents[17]. These preclinical achievements serve as the basis of clinical trials in which PAL is being combined with chemotherapy (NCT02501902) and targeted therapy (e.g., mTOR inhibitor, NCT03065062; extracellular signal–regulated kinase inhibitor, NCT03454035). However, separate administration of paired agents may overlook the role of biodistribution, inadequate tumor access, independent pharmacokinetics (PKs), and intratumoral drug ratio, which are the key factors for drug synergy in vivo. Recently, co-delivery nanoparticles (NPs) with ratiometric designs have provided a practical approach for the in vivo delivery of drug combination, with the purpose to support optimal drug synergy at the tumor site[18,19]. In 2017, the Food and Drug Administration (FDA) approved the first ratiometric liposome (Vyxeos®) for the co-delivery of daunorubicin/cytarabine, capable of controlling drug ratios post intravenous (i.v.) injection, for acute myeloid leukemia (www.accessdata.fda.gov/drugsatfda_docs/label/2017/209401s000lbl.pdf). The field of cancer nanomedicine is now well positioned for ratiometric co-delivery formulations that can effectively implement versatile drug combinations[19,20].

In light of the above information, we propose the ratiometric co-delivery nanocarrier for CDK4/6i combination therapy in PDAC. In this study, we aim to deliver CDK4/6i PAL and autophagy inhibitor hydroxychloroquine sulfate (HCQ) considering both the aspects of cancer biology and chemical structure of active pharmaceutical ingredient (API). PAL/HCQ have been chosen given that CDK4/6i may induce cancer cell autophagy as a stress-tolerance response, evidenced by upregulation of various autophagic markers via a reactive oxygen species (ROS)-mediated mechanism[13]. The use of autophagy inhibitors in combination with CDK4/6i has demonstrated promising synergy in multiple cancer types, such as breast cancer, gastric cancer, and PDAC[13,21]. Equally important, both APIs are amphiphilic weak base with protonatable amine groups and similar pKa values[22,23], allowing efficient drug import into a tailor-designed lipid-coated

mesoporous silica NP (MSNP) platform that has been established in our laboratory[24–26]. Based on in vitro screening and software-mediated synergy study, we design the drug ratio in PAL/HCQ-laden nanocarrier that exhibits optimized level of synergism. We also demonstrate that the employment of ratiometric co-delivery is critical to achieve synchronized PK and intratumoral biodistribution of PAL/HCQ, which can otherwise be distributed independently. Animal studies were performed in both subcutaneous and orthotopic PANC-1 models, demonstrating that ratiometric co-delivery nanocarrier lead to potent anticancer effect compared to various controls, including the free drug pair. In addition to the effective tumor shrinkage, co-targeting CDK4/6 and autophagy results in the activation of Bcl anti-apoptotic pathway post multiple i.v. injections. This has prompted us to introduce a Bcl inhibitor to generate further enhanced efficacy in PDAC model both in vitro and in vivo.

## Results

**Free PAL/HCQ induces synergistic growth arrest in PDAC cells.** To design an optimal co-delivery nanoformulation, our first attempt was to validate the synergism and determine the range of synergistic ratio between PAL and HCQ in PDAC cells. We performed 5-bromo-2-deoxyuridine (BrdU) assay in PANC-1 cells treated by free PAL/HCQ mixture with drug ratios over the range of 10:1–1:20 (molar ratio) (Fig. 1a). The drug concentration in the mixture that inhibited 50% cell proliferation (IC50) was compared to the mono-treatment of PAL and HCQ. IC50 of free PAL alone and HCQ alone in PANC-1 cells was determined as 8.1 μM and >50 μM, respectively. Significantly reduced IC50 was observed when the combination of PAL/HCQ was applied in a ratio-dependent fashion. For example, at PAL/HCQ ratio of 1:5, the IC50 of PAL was reduced from 8.1 to 1.1 μM in PANC-1 cells compared to mono-treatment. To quantitatively display the synergistic effect, combination index (CI) was calculated via the CompuSyn software[27]. CI < 1 indicates a synergistic effect and a strong level of synergy if CI ≤ 0.5. In PANC-1 cells, PAL/HCQ pair displayed strong level of synergy (as indicated by CI < 0.3) over broad range of PAL/HCQ molar ratio from 1:1 to 1:20. In addition to PANC-1 cells, 4 PDAC cell lines were tested, including MIA PaCa-2, BxPC-3, HPAF-II, and AsPC-1 cells (Fig. 1a). All the PDAC cells were positive for phospho-Rb (pRb)/Rb expression (Supplementary Fig. 1). The CI analysis suggested that free PAL/HCQ mixture of 1:5 (molar ratio) led to strong synergy in all the five PDAC cell lines. Detailed results for in vitro screening were provided in Supplementary Fig. 2. Moreover, crystal violet staining, which provided direct visualization to compare the differences in cell proliferation, further confirmed the effective growth inhibition by PAL/HCQ pair at a synergistic molar ratio (1:5) in PANC-1 cells (Fig. 1b).

Before moving into the study of nanocarrier design, we performed a series of cell-based assays to investigate the associated biological events, such as cell proliferation, cell cycle, and autophagy flux. We demonstrated that free PAL/HCQ mixture (1:5 molar ratio) induced the most significant down-regulation of Ki67 (marker for proliferation) in PANC-1 cells compared to PAL or HCQ mono-treatment (Fig. 1c). Meanwhile, >90% of PAL/HCQ-treated PANC-1 cells were subjected to G1 cell cycle arrest (Supplementary Fig. 3), which was accompanied by efficient inhibition of pRb (Fig. 1d). We also looked at autophagy blockade using a LC3B-GFP-RFP dual reporter assay (Fig. 1e), in which PANC-1 cells were transfected to express GFP-LC3B (acid-sensitive) and RFP-LC3B (acid-insensitive) to identify autophagic vacuoles (Fig. 1e, upper panel)[28]. The data indicated that PAL mono-treatment induced accelerated autophagy flux in PANC-1 cells without interference on

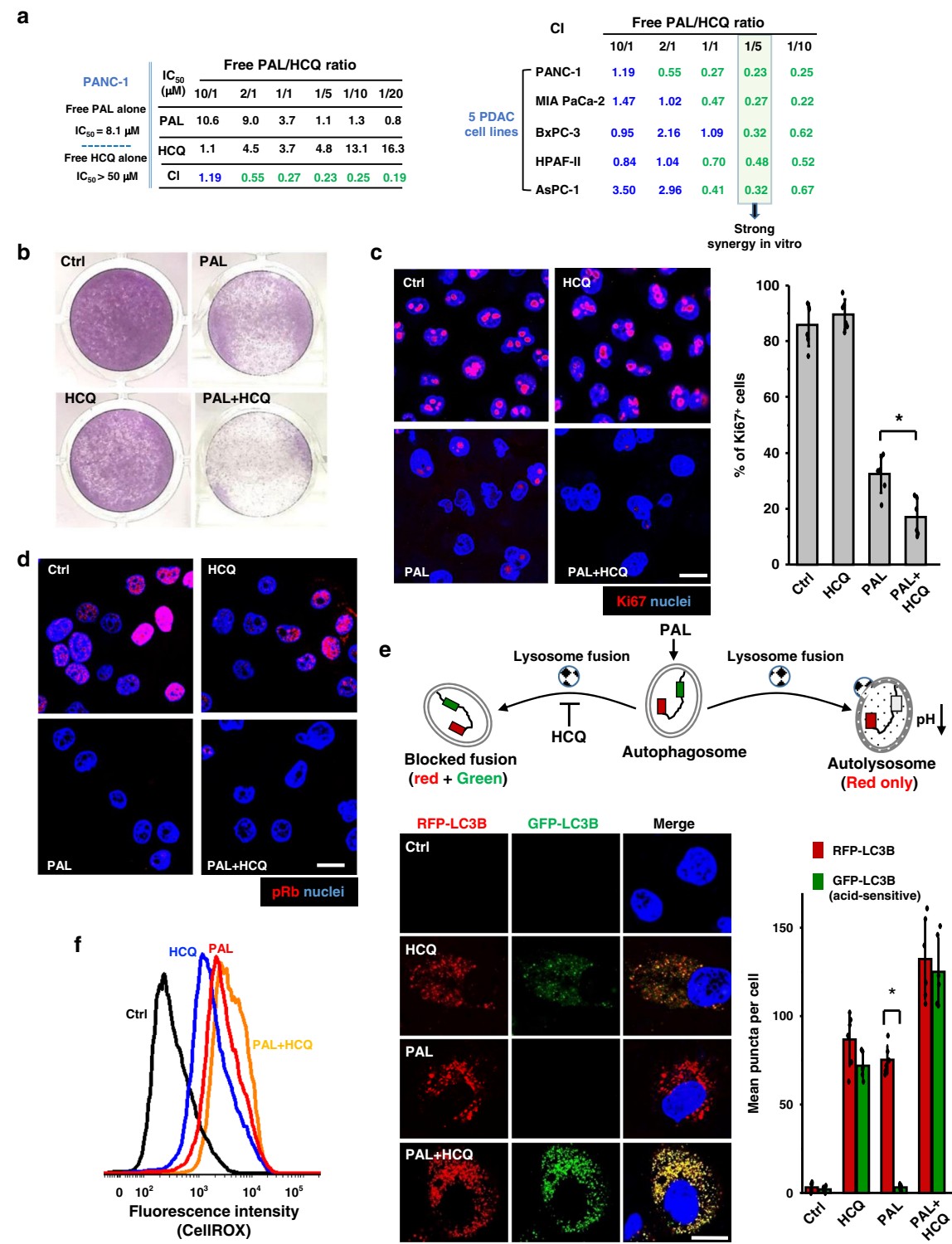

**a**

**PANC-1**

| IC$_{50}$ (µM) | Free PAL/HCQ ratio | | | | | |
|---|---|---|---|---|---|---|
| | 10/1 | 2/1 | 1/1 | 1/5 | 1/10 | 1/20 |
| PAL | 10.6 | 9.0 | 3.7 | 1.1 | 1.3 | 0.8 |
| HCQ | 1.1 | 4.5 | 3.7 | 4.8 | 13.1 | 16.3 |
| CI | 1.19 | 0.55 | 0.27 | 0.23 | 0.25 | 0.19 |

Free PAL alone IC$_{50}$ = 8.1 µM

Free HCQ alone IC$_{50}$ > 50 µM

| CI | Free PAL/HCQ ratio | | | | |
|---|---|---|---|---|---|
| | 10/1 | 2/1 | 1/1 | 1/5 | 1/10 |
| PANC-1 | 1.19 | 0.55 | 0.27 | 0.23 | 0.25 |
| MIA PaCa-2 | 1.47 | 1.02 | 0.47 | 0.27 | 0.22 |
| BxPC-3 | 0.95 | 2.16 | 1.09 | 0.32 | 0.62 |
| HPAF-II | 0.84 | 1.04 | 0.70 | 0.48 | 0.52 |
| AsPC-1 | 3.50 | 2.96 | 0.41 | 0.32 | 0.67 |

5 PDAC cell lines

Strong synergy in vitro

**b** Ctrl, PAL, HCQ, PAL+HCQ

**c** Ctrl, HCQ, PAL, PAL+HCQ — Ki67 nuclei; % of Ki67⁺ cells

**d** Ctrl, HCQ, PAL, PAL+HCQ — pRb nuclei

**e** PAL — Lysosome fusion; Blocked fusion (red + Green); Autophagosome; Autolysosome (Red only); HCQ; pH↓

RFP-LC3B, GFP-LC3B, Merge — Ctrl, HCQ, PAL, PAL+HCQ

RFP-LC3B; GFP-LC3B (acid-sensitive); Mean puncta per cell

**f** Ctrl, HCQ, PAL, PAL+HCQ — Fluorescence intensity (CellROX)

autophagosome–lysosome fusion, evidenced by the presence of autolysosomes (RFP-only puncta) and near-absence of autophagosomes (both GFP and RFP puncta). This contrasted with cells treated by PAL/HCQ pair, in which the highest level of autophagosome accumulation was observed among all the treatment groups. The result can be attributed to the enhanced autophagy influx induced by PAL but impaired autophagy efflux due to the blockade on autophagosome–lysosome fusion by HCQ[13]. Since ROS consumption is generally regarded as the requisite for autophagy flux[13], we further examined the intracellular ROS level via CellROX staining, followed by flow

cytometric analysis (Fig. 1f). Significantly enhanced ROS accumulation in PAL/HCQ-treated PANC-1 cells was observed compared to PAL mono-treatment, demonstrating the pharmacological interruption of autophagy.

In addition to PANC-1 cells, the synergism between PAL and HCQ was demonstrated in MIA PaCa-2 cells. Consistent with the results in PANC-1 cells, PAL/HCQ mixture led to similar response in anti-proliferation, pRb inhibition, autophagy regulation, and ROS induction (Supplementary Fig. 4). These results collectively suggested that PAL mono-treatment induced an ROS-mediated autophagy flux to circumvent the anti-proliferative

**Fig. 1 PAL/HCQ free drug combination led to synergized anti-proliferative effect in PDAC cell lines. a** BrdU proliferation assay was performed on PDAC cells treated by free PAL/HCQ mixtures at various combination molar ratios. Treatment of each concentration was tested in triplicate by a 96-well BrdU proliferation assay ($n = 3$). Combination index (CI) were calculated for each combination ratio via the CompuSyn software. CI < 1 indicated a synergistic effect and CI ≤ 0.5 indicated a strong level of synergy. The detailed in vitro screening data in five cell lines are provided online (Supplementary Fig. 2). PANC-1 cells were incubated with PAL/HCQ combination or single drug at equivalent concentration (PAL/HCQ = 1:5, PAL = 2.5 μM, HCQ = 12.5 μM) for 72 h and subjected to the following assays to elaborate the associated biological events. **b** Crystal violet staining of PANC-1 cells visualized the synergistic anti-proliferative effect by PAL/HCQ pair. **c** Immunofluorescence staining of Ki67 proliferative marker (red), followed by quantification of Ki67+cells in six randomly selected ROIs ($n = 6$). Free PAL/HCQ combination displayed significantly enhanced anti-proliferative effect compared to PAL mono-treatment ($p = 0.0027$). Scale bar: 20 μm. **d** Immunofluorescence staining of phospho-Rb (pRb) suggested that PAL/HCQ combination effectively reduced the pRb expression, which was the hallmark for CDK4/6i-mediated growth arrest. The experiment was repeated twice. Scale bar: 20 μm. e GFP-RFP-LC3B dual reporter assay to determine the blockade on autophagosome–lysosome fusion. Six randomly selected ROIs were used in the analysis ($n = 6$). The abundance of RFP-LC3B puncta was significantly enhanced over GFP puncta in PAL-treated cells ($p = 2.45 \times 10^{-8}$), demonstrating the accumulation of autolysosomes. HCQ in combination with PAL prevented the autophagosome–lysosome fusion. Scale bar: 20 μm. **f** The generation of reactive oxygen species (ROS) were stained by CellROX reagent, and the mean intracellular ROS level was analyzed via flow cytometry. Data represent mean ± SD, and the statistical difference was evaluated by one-way ANOVA followed by Tukey's post hoc test (*$p < 0.05$). Source data are provided as a Source data file.

effect in PDAC cells, and simultaneous blockade of autophagy resulted in synergistic PDAC inhibition. We then used the information from the free drug combination as a reference to perform the custom-designed loading of PAL/HCQ pair in a ratiometric co-delivery nanocarrier.

**Development of ratiometric co-delivery nanocarrier for PAL/HCQ.** We have previously demonstrated the utilization of multifunctional lipid bilayer (LB)-coated MSNP (silicasome) for the delivery of chemotherapeutics[24–26,29]. This includes the encapsulation of a hydrophilic drug such as gemcitabine into the porous space in MSNP (core), with incorporation of a hydrophobic drug such as paclitaxel into the LB coating (shell)[29]. Another approach is to employ remote loading of single payload, exemplified by the delivery of irinotecan[24–26]. While we have considered the core/shell co-delivery approach for PAL/HCQ, the chemical structure of both APIs precludes the possibility of lipid incorporation, which usually provides limited loading capacity (0–3 wt%) and narrow tunability of drug ratios for co-packaging. Since PAL and HCQ are amphipathic and protonatable weak base, with comparable solubility in water (>5 mg/mL), similar range of pKa (PAL: pKa = 7.4; HCQ: pKa$_1$ = 9.7, pKa$_2$ = 8.3)[22,23] and molecular weights (PAL: 447.5 Da; HCQ: 335.9 Da), we asked whether it was possible to ratiometrically co-encapsulate PAL and HCQ into the porous space of MSNP via simultaneous remote loading.

The remote co-encapsulation of PAL/HCQ pair was illustrated in Fig. 2a. Our attempts began from identification of API that meets certain physiochemical criteria, such as the presence of amine group(s), pKa, water solubility, octanol/water partition coefficient, molecular weight, etc. The preference is weak base that can be efficiently imported into the nanocarrier via a proton-releasing trapping agent. Compared to the import of a single drug, co-package is more complex and requires additional considerations on the choice of trapping agent (that is ideally compatible for both APIs), feed ratio, loading condition, etc. Principles for formulation design was summarized in Supplementary Fig. 5. Accordingly, a list of parameters, such as pH, MSNP/PAL ratio, MSNP/HCQ ratio, PAL/HCQ feed ratio, incubation time, etc., were iteratively optimized. The preferred loading condition is outlined in Supplementary Fig. 6.

The co-delivery nanocarriers, consisting of an ~58-nm MSNP core with LB coating 1,2-distearoyl-sn-glycero-3-phosphocholine (DSPC)/cholesterol (Chol)/1,2-distearoyl-snglycero-3-phosphoethanolamine-N-[methoxy(polyethylene glycol)-2000] (ammonium salt) (DSPE-PEG$_{2000}$) = 3:2:0.15, molar ratio)[24–26], were first encapsulated with triethylammonium sucrose octasulfate (TEA$_8$-SOS) at 80 mM. The pre-encapsulation of TEA$_8$SOS created a

pH < 5.5[30] inside of MSNP. The presence of this pH gradient has allowed the co-import of amphipathic drugs across the LB into the interior of the porous structures, where the protons released by the entrapped TEA$_8$SOS convert the drug pair into their protonated counterparts with enhanced hydrophilicity. The anionized sucrose octasulfate further precipitated the drug pair and prevented them from diffusion out of the packaging space (Fig. 2a). A series of PAL/HCQ co-delivery nanoformulations were prepared and the physiochemical properties were characterized (Fig. 2b). High-performance liquid chromatography (HPLC) or ultraviolet (UV) detection were performed to determine the drug-loading content. Across different feed ratios of PAL/HCQ, the total loaded drug accounted for 15–20 wt% of the NPs. Feed molar ratios of PAL/HCQ between 2:1 and 1:7 has allowed us to control the measured loading ratios of PAL/HCQ between 2.4:1 and 1:6 in the NPs (Fig. 2b). Drug co-import beyond this range turned out to be difficult due to the significantly reduced colloidal stability of drug-laden particles, as well as the prevailing transmembrane diffusion of the single drug.

To validate the drug synergism by different co-delivery nanoformulations, we performed BrdU assay on PANC-1 cells to examine IC$_{50}$ and CI values (Fig. 2c). Similar to the study on free drug combinations, synergism was detected from all tested co-delivery NPs. We chose the co-delivery nanoformulation with PAL/HCQ feed molar ratio of 1:5 (with measured loading ratio of 1:4.3) for the following study because it exhibited the highest level of synergism (CI = 0.24, arrow in Fig. 2c). This formulation (abbreviated as "co-delivery NP") led to significant reduction in IC$_{50}$ compared to single drug-laden NPs, i.e., 4-fold more efficient than PAL-only NP (PAL NP) and >10-fold more efficient than HCQ-only NP (HCQ NP) (Fig. 2c).

The physiochemical characterization of the optimized PAL/HCQ co-delivery NP is summarized in Fig. 2d. The primary size and morphology of co-delivery NP was characterized with cryogenic electron microscopy (cryoEM; Fig. 2d), which suggested a highly uniform particle size of ~72 nm with intact LB coating (~7-nm thick). Co-delivery NP exhibited a hydrodynamic size of ~126 nm and zeta potential of ~−10 mV in saline.

To confirm the pharmacological impact on cell proliferation, cell cycle progression, and autophagy, co-delivery NP-treated PANC-1 cells were investigated with the in vitro assays, similar to Fig. 1. Consistent with the data from the free drug combination, co-delivery NP demonstrated the most efficient anti-proliferative effect (Fig. 2e) by downregulation of Ki67 (Fig. 2f), compared to single drug-laden NPs. Distinct G1 cell cycle arrest (Fig. 2g) and pRb inhibition (Fig. 2h) were also observed. The raw data of in vitro synergy are provided in Supplementary Fig. 7A–C.

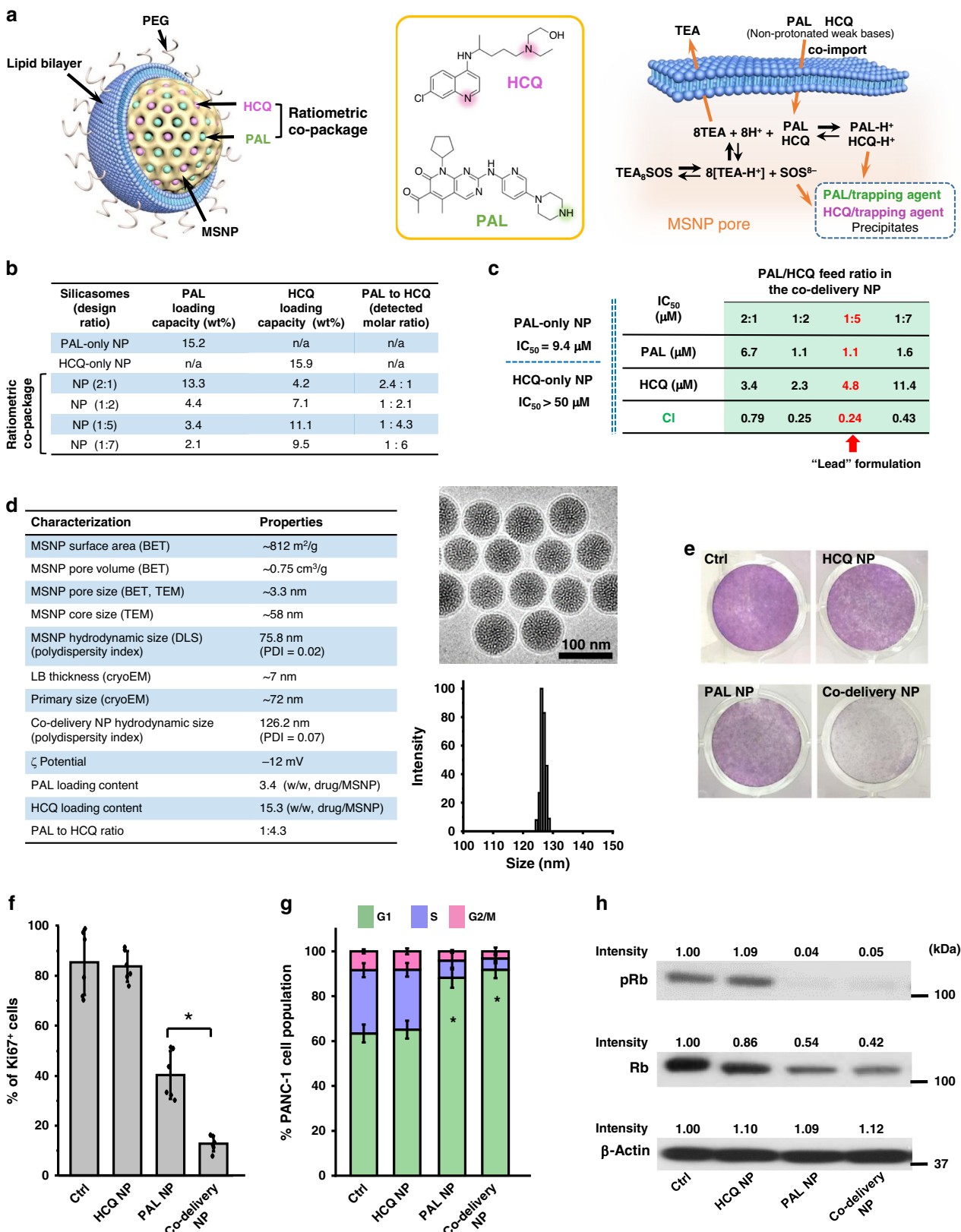

**Fig. 2** (Continued)

For co-delivery NP, we investigated PAL-induced autophagy and HCQ-mediated concurrent blockade in detail. Transmission electron microscopic (TEM) examination of autophagic vacuoles in co-delivery NP-treated PANC-1 cells demonstrated the most distinct accumulation of autophagosomes (accelerated efflux with impaired autophagosome–lysosome fusion), which were characterized by the presence of bilayered vacuoles with an electron-lucent interior (arrows in Fig. 2i)[31]. This contrasted with PAL

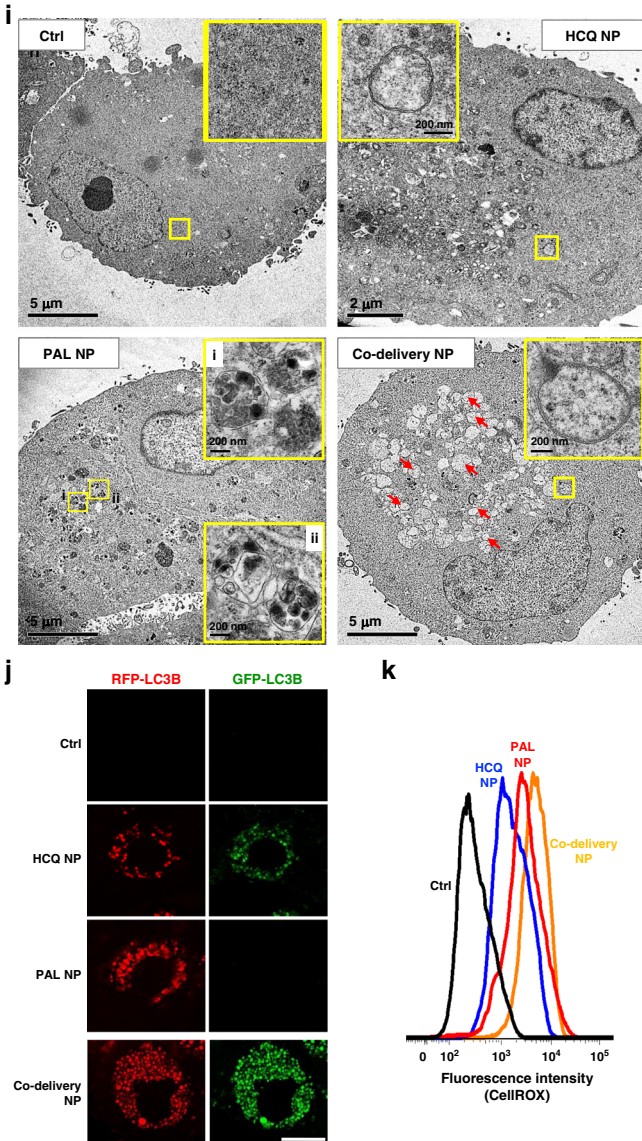

**Fig. 2 Development of ratiometric nanoformulation for the co-delivery of PAL and HCQ. a** Schemes depicting the structure of lipid-coated MSNP nanocarrier and the mechanism for simultaneous remote loading of PAL and HCQ using a trapping agent. $TEA_8SOS$-laden particles were incubated with the solution of PAL and HCQ mixed at predefined molar ratios, allowing the amphipathic drugs to diffuse across the lipid bilayer for protonation by $TEA_8SOS$. **b** Preparation of co-delivery nanocarriers with various PAL/HCQ ratios. **c** BrdU proliferation assay to calculate $IC_{50}$ and CI for co-delivery nanoparticles. Each concentration was tested in triplicate ($n = 3$). Co-delivery formulation with PAL/HCQ feed molar ratio of 1:5 (loading ratio of 1:4.3) displayed the best level of synergy and has been chosen for the following study (red arrow). **d** Summary of physiochemical properties of co-delivery NP. CryoEM of co-delivery NP suggested intact lipid bilayer coating and uniform size distribution. Scale bar: 100 nm. **e** Crystal violet staining to visualize the growth-inhibitory effect in PANC-1 cells incubated with co-delivery NP for 72 h with equivalent concentration (PAL = 2.5 μM, HCQ = 10.8 μM). **f** Co-delivery NP significantly improved the anti-proliferative effect compared to PAL NP ($p = 5.7 \times 10^{-4}$). Six ROIs were analyzed for the Ki67 expression ($n = 6$). **g** Cell cycle analysis using flow cytometry (triplicate). Both PAL NP and co-delivery NP induced significant G1 cell cycle arrest compared to control with $p$ values of $3.3 \times 10^{-4}$ and $1.2 \times 10^{-4}$, respectively. **h** Western blot to reflect the impact on the expression of phospho-Rb and total Rb. **i** Representative TEM characterization of autophagic vacuoles in PANC-1 cells. **j** GFP-RFP-LC3B reporter assay ($n = 6$ ROIs) to visualize and compare the intracellular level of autophagosomes (GFP and RFP puncta) and autolysosomes (RFP-only puncta). Scale bar: 20 μm. Quantification of GFP-RFP-LC3B is also shown in Supplementary Fig. 7D. **k** ROS production probed by CellROX staining suggested that the autophagy blockade by co-delivery NP was ROS mediated. One of the two independent repetitions with similar results is shown here for **b**–**k**. Data represent mean ± SD; statistical difference was evaluated by one-way ANOVA followed by Tukey's post hoc test (*$p < 0.05$). Source data are provided as a Source Data file.

NP-treated PANC-1 cells, in which abundance of autolysosomes were observed (accelerated and intact autophagy flux), typified by the single-membraned vacuolar structure that enveloped partially degraded, electron-dense cellular components (zoom-in regions i and ii in Fig. 2j)[31]. Moreover, the accumulation of autophagosomes by co-delivery NP was also confirmed in GFP-RFP-LC3B dual reporter assay (Fig. 2j and Supplementary Fig. 7D). Meanwhile, significant blockade of p62 degradation (Supplementary Fig. 7E) and interruption on ROS consumption (Fig. 2k and Supplementary Fig. 8) corroborated that co-delivery

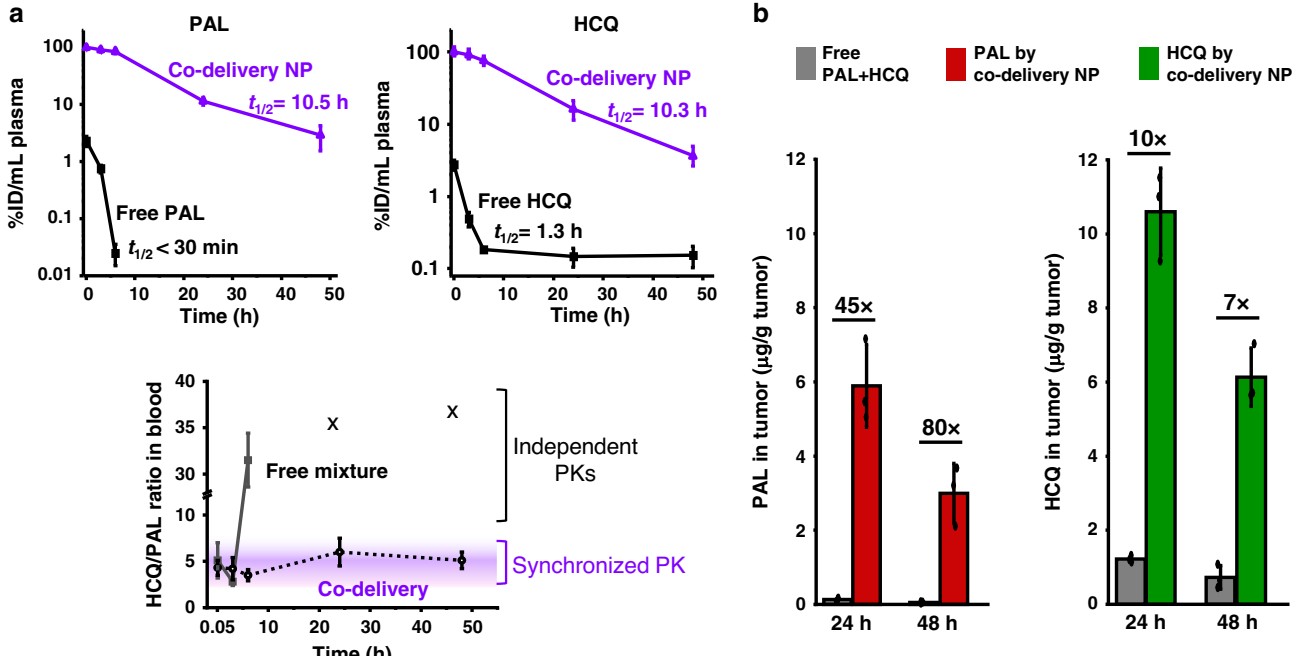

**Fig. 3 Co-delivery NP significantly enhanced the synchronized PK and intratumoral distribution of PAL/HCQ pair after i.v. administration in subcutaneous PANC-1 xenograft mice model.** Mice received single i.v. injection of co-delivery NP (with the particle dose of 210 mg/kg) or free PAL/HCQ pair, with an equivalent PAL dose of 10 mg/kg and an HCQ dose of 33 mg/kg that correlated to a PAL/HCQ molar ratio of 1:4.3. **a** Evaluation of PK profile and PAL/HCQ ratio in plasma. Plasma was collected 0.083, 3, 6, 24, and 48 h post i.v. injection ($n = 3$ animals included in each data point). The plasma content of PAL and HCQ was determined by UPLC-MS and expressed as percentage of total injected dose (% ID) per mL of plasma. The circulatory half-life ($t_{1/2}$) were calculated by the PKSolver software. The PAL/HCQ ratio in plasma were calculated and plotted versus time. **b** PAL/HCQ content at the tumor site. In a separate experiment, mice were sacrificed at 24 or 48 h post i.v. injection ($n = 3$ animals included in each data point). PAL/HCQ content in tumor site was determined by UPLC-MS, and comparison was made between co-delivery NP and direct administration of free PAL/HCQ mixture. Data were obtained from one set of independent experiment without repetition. Data represent mean ± SD; statistical difference was evaluated by one-way ANOVA followed by Tukey's post hoc test. Source data are provided as a Source data file.

NP can efficiently inhibit PAL-induced autophagy. In addition, western blot detection of LC3B and p62 confirmed that co-delivery NP significantly inhibited the autophagy influx compared to PAL NP alone, as co-delivery NP led to enhanced LC3-II/LC3-I ratio and reduced p62 degradation (Supplementary Fig. 9).

Taken together, co-delivery NP exhibited good competency to support the in vitro synergy between PAL/HCQ over a wide range of ratiometric designs. The underlying biological events that led to a synergistic anti-proliferative effect from the co-delivery NP was consistent with our prior demonstration with the free drug pair.

**Enhanced PK and intratumoral drug content by co-delivery NP.** Next, we explored the PK profile and the intratumoral content of PAL/HCQ in a PANC-1 xenograft model. Subcutaneous PANC-1 xenograft model was established on female athymic Balb/c nude mice. With an average tumor size of ~10 mm, PANC-1 tumor-bearing mice received a single i.v. injection of co-delivery NP (210 mg NP/kg) with a PAL/HCQ loading molar ratio of 1:4.3, correlating to a PAL dose of 10 mg/kg and an HCQ dose of 33 mg/kg. Mice receiving a single i.v. injection of free PAL/HCQ mixture at equivalent dose were also compared. Blood samples were collected at the indicated time points (0.083, 3, 6, 24, and 48 h) for the PK study. Mice were euthanized 24 or 48 h ($n = 3$ for each time point) post i.v. injection to assess the intratumoral drug content. PAL and HCQ concentration in plasma was detected via ultra-performance liquid chromatography–mass spectrometry (UPLC-MS; Supplementary Fig. 10) and expressed as percentage

of injected dose per unit volume of plasma (%ID/mL). PK profile of PAL and HCQ was assessed by the PKSolver software (Fig. 3a), and a comparison was made for each drug between administration via the co-delivery NP or the free drug mixture, respectively. In the case of co-delivery NP, the concentration of both PAL and HCQ were maintained at >10%ID/mL in plasma over 24 h post injection. Co-delivery NP significantly enhanced the circulatory half-life ($t_{1/2}$) of PAL to 10.5 h compared to <30 min for free PAL, and simultaneously prolonged the $t_{1/2}$ of HCQ to 10.3 h compared to 1.3 h for free HCQ (Fig. 3a). In the case of the free PAL/HCQ mixture, departure from proportionality was observed as early as 6 h post injection (Fig. 3a). Twenty-four and 48 h post injection, free PAL diminished beyond the detection limit, and <0.2%ID of free HCQ was still identified in plasma, resulting in an undetectable PAL/HCQ ratio. The result correlated to the unsynchronized PK of the free single drug as previously speculated[32–34]. On the contrary, consistent PAL/HCQ ratio by co-delivery NP was maintained in the plasma over the entire time course of the study (Fig. 3a, purple zone). Slight fluctuation was detected between PAL/HCQ = 1:5.9 and 1:3.5 (molar ratio), but all the PAL/HCQ ratios in plasma fell into the synergistic range that we demonstrated in vitro (Figs. 1a and 2c). In addition, intratumoral drug concentration was also determined via UPLC-MS. Twenty-four and 48 h post injection, co-delivery NP led to a 45–80-fold augmented intratumoral PAL content and a 7–10-fold augmented HCQ content simultaneously, compared to free drug mixture (Fig. 3b). In the clinics, PAL is administered for 21 consecutive days (75–125 mg/patient) followed by 7 days off per treatment cycle (https://www.accessdata.fda.gov/drugsatfda_docs/label/2017/207103s004lbl.pdf), and HCQ is typically given

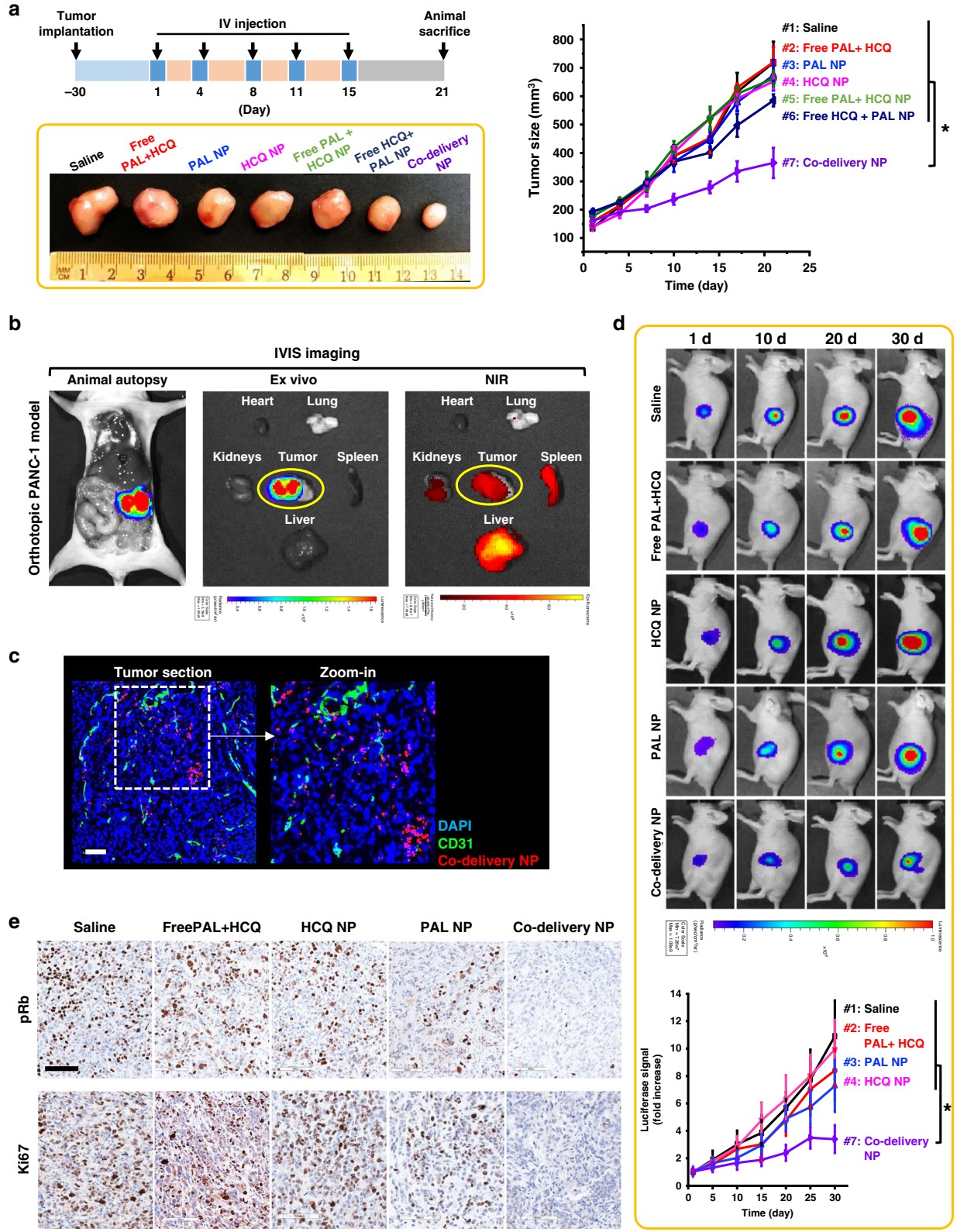

200–800 mg/patient over weeks[35]. The results demonstrated that co-delivery NP can effectively prolong the drug exposure, which is essential to dose reduction. Furthermore, co-delivery NP provided synchronized PK and drug ratio, which has prompted us to investigate the therapeutic efficacy in PDAC animal models.

**Synergistic antitumor effects by co-delivery NP.** For proof of principle, we first performed the efficacy study in a subcutaneous PANC-1 xenograft model. As indicated in the treatment scheme (Fig. 4a), PANC-1 tumor-bearing mice received 5 i.v. injections of co-delivery NP at particle dose of 210 mg/kg/injection (PAL: 10

**Fig. 4 Co-delivery NP demonstrated synergistic antitumor effects in both subcutaneous and orthotopic PANC-1 models. a** Efficacy study in subcutaneous PANC-1 xenograft model. Tumor bearing mice ($n = 4$) received 5 i.v. injections of co-delivery NP (210 mg particles/kg/injection), as well as single drug-laden nanoparticle or free drugs at an equivalent PAL dose of 10 mg/kg/injection and an HCQ dose of 33 mg/kg/injection. Mice treated with saline were also studied as control. The tumor size was monitored for 21 days from the first injection and plotted versus time. Co-delivery NP led to the most potent tumor inhibition compared to all the controls, e.g., co-delivery NP versus Free PAL + HCQ ($p = 7 \times 10^{-5}$). Representative images of tumor harvested on day 21 further demonstrated the significantly enhanced antitumor efficacy by co-delivery NP. **b** Confirmative biodistribution study in orthotopic PANC-1 xenograft model. Mice ($n = 3$) received single i.v. injection of DyLight680-labeled co-delivery NP (210 mg particles/kg) and sacrificed 48 h post injection. The orthotopic tumor site was localized by bioluminescence-based imaging. Ex vivo IVIS imaging were performed to determine the NIR signals from major organs and tumor to identify the distribution of nanoparticles. **c** Confocal microscopy confirmed the abundance of DyLight680-labeled co-delivery NP (red) in the section of orthotopic tumor stained with anti-CD31 antibody (green) and DAPI (blue). Scale bar: 100 μm. **d** Efficacy study in orthotopic PANC-1 model. Mice ($n = 4$) received the same dosage regimen as in **a**. Live-animal IVIS imaging was performed to monitor the orthotopic tumor growth for 30 days from the first injection and the representative bioluminescence is shown. The bioluminescence intensity was quantified at the ROI by the IVIS Living Image software v4.5 and plotted versus time. Co-delivery NP outperformed all the control groups, e.g., co-delivery NP versus Free PAL + HCQ ($p = 0.025$). **e** Immunohistochemical staining of tumor sections harvested on day 30 revealed significant inhibition of pRb and Ki67. Scale bar: 100 μm. Data were obtained from one experiment without repetition. Data represent mean ± SD; statistical difference was evaluated by one-way ANOVA followed by Tukey's post hoc test (*$p < 0.05$). Source data are provided as a Source data file.

mg/kg/injection; HCQ: 33 mg/kg/injection). Free PAL/HCQ mixture with equivalent dose of PAL and HCQ were also studied. Other groups include single drug-laden NPs (PAL-only; HCQ-only), free single drug combined with single drug-laden NP (free PAL + HCQ-only NP; free HCQ + PAL-only NP), and saline control. Primary tumor size was monitored over the treatment schedule and plotted versus time (Fig. 4a). Tumor growth inhibition was observed in co-delivery NP-treated group (group #7 in Fig. 4a), with statistical significance compared to other groups ($p < 0.05$). Moderate retardation of tumor growth was observed in the group of free HCQ + PAL-only NP (group #6), potentially due to the presence of persistent but residual HCQ in circulation as well as the tumor site (as demonstrated in Fig. 3) that may synergize with PAL-only NP. No obvious antitumor effect was observed for the rest of the treatment groups, including the free drug mixture (group #2). Photograph of the representative tumor tissues from each treatment group is shown in the inserted box in Fig. 4a.

It is generally agreed that PDAC therapies should be tested in stringent models that mimic human diseases[36]. Confirmative studies of biodistribution and efficacy were also performed in orthotopic PANC-1 model. To visualize the progression of orthotopic tumors in mice, PANC-1 cells stably transfected with luciferase reporter[29] were used for tumor inoculation to facilitate bioluminescence-based IVIS imaging. In the biodistribution study, orthotopic PANC-1 tumor-bearing mice received single i.v. injection of near infrared (NIR) fluorophore (DyLight680)-labeled co-delivery NP at a particle dose of 210 mg/kg. Forty-eight hours post injection, animals were intraperitoneally (i.p.) injected with D-luciferin (75 mg/kg) to localize orthotopic tumor site, before ex vivo imaging was performed (Fig. 4b). Abundant distribution of co-delivery NP at the tumor site was evidenced by IVIS detection of NIR fluorescence, which overlapped consistently with the bioluminescence signal from the tumor site (Fig. 4b). Confocal microscopic study with tumor section (Fig. 4c) confirmed the intratumoral abundance of co-delivery NP. The immunofluorescence staining revealed a heterogeneous distribution profile, with a relatively high particle density in the vicinity of the CD31$^+$ tumor blood vessels post single i.v. injection. This is compatible with our previous demonstration of the micro-heterogeneity of NP distribution in BxPC-3- and KPC-derived PDAC models[24,37]. Our results suggested that the micro-heterogeneity could be alleviated after multiple i.v. injections, which can be reflected in the antitumor efficacy data (Fig. 4d). In addition to the primary tumor site, co-delivery NP were also identified in the liver and spleen and sparingly in the kidneys.

Next, confirmative efficacy study was performed in the orthotopic PANC-1 model with essential control groups. Mice received i.v. injections following the same treatment regimen as in the subcutaneous model. Bioluminescence-based IVIS monitoring of tumor development was performed for up to 30 days (Fig. 4d and Supplementary Fig. 11). Consistent therapeutic potency was demonstrated for co-delivery NP (treatment #7 in Fig. 4d), which exhibited significantly advantageous growth inhibition on orthotopic PANC-1 tumor compared to single drug-laden NPs or free drug mixture at identical dose. In the clinics, the major adverse events associated with conventional PAL therapy include neutropenia and leukopenia, which led to the discontinuation of treatment[1]. To address this, complete blood count and chemistry assays were performed to dissect any potential adverse effects. Co-delivery NP showed no significant change in blood counts, red blood cell differentiation, or white blood cell differentiation (Supplementary Fig. 12A–C). No major abnormalities have been identified in blood chemistry (Supplementary Fig. 12D) and histological assessment (Supplementary Fig. 12E).

**Repetitive dose of co-delivery NP activates anti-apoptotic pathway.** To interpret the efficacy of co-delivery NP, a list of histological assessments was performed. There was an inhibitory effect on pRb as well as Ki67 in tumor receiving co-delivery NP (Fig. 4e). Immunohistochemical (IHC) staining of pRb and Ki67 at low magnification also allowed us to evaluate the anti-proliferative effect regarding the intratumoral heterogeneity (Supplementary Fig. 13). Co-delivery NP led to relatively uniform anti-proliferative effect, presumably due to the enhanced biodistribution profile and repetitive i.v. injections. An important caveat is that if a heterogeneous distribution profile of NPs becomes a major concern in the future, we will contemplate the use of two-wave therapy to improve intra-PDAC distribution of NPs, by nano-enabled TGF-β inhibition that decreases pericyte coverage of the PDAC vasculature[37].

In addition to Ki67 and pRb, we also investigated cleaved caspase-3 (CC-3), as the biomarker for apoptosis. While effective in vivo, Fig. 5a showed a low CC-3 expression in the co-delivery NP-treated group at the endpoint of efficacy study. The additional CC-3 staining was provided in Supplementary Fig. 14. Similar findings are shown in vitro, demonstrating the absence of CC-3 in co-delivery NP-treated PANC-1 cells (Fig. 5a, lower panel). This differed from the conventional chemotherapeutics, such as paclitaxel, which elicited strong CC-3 expression in PANC-1 model[38]. The mechanism of PAL treatment beyond inhibition of cell cycle progression is complex and yet to be fully

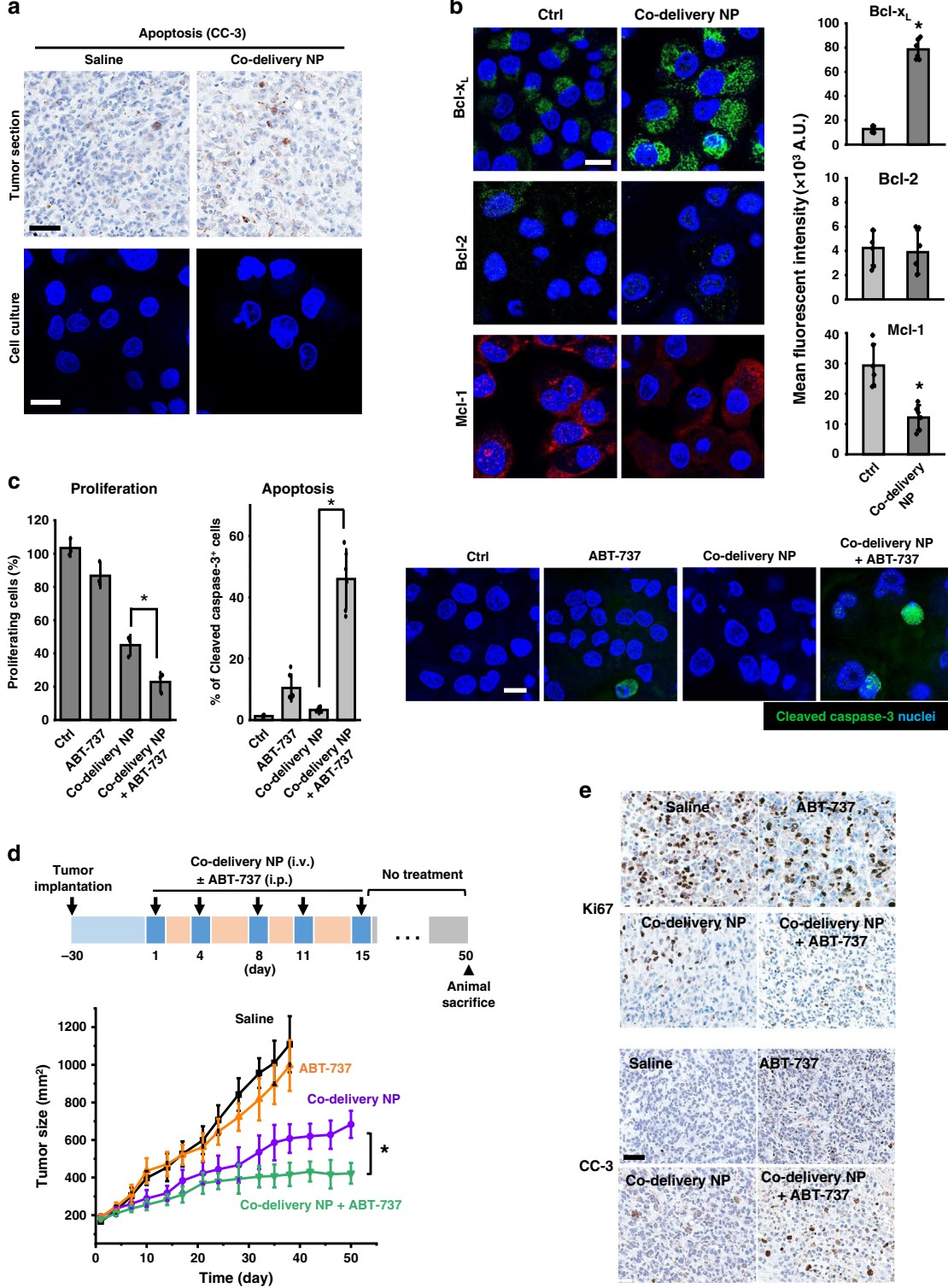

understood[39,40]. Emerging evidence suggests that anti-apoptotic proteins may serve as a key defense system for CDK*i*. Prior demonstration includes upregulation of Bcl-2 family protein in alvocidib-treated large B-cell lymphoma[41] and voruciclib-treated leukemia[42], as well as Bcl-2 overexpression in ER$^+$ breast cancer[43,44]. We then explored a series of Bcl-2 subfamily proteins (Bcl-x$_L$, Bcl-2, Mcl-1)[45] in PANC-1 cells receiving co-delivery NP. While no significant change of Bcl-2 was observed, the results suggested significantly elevated sequestration of Bcl-x$_L$ in vitro (Fig. 5b). In addition, Mcl-1 degradation was also evidenced

(Fig. 5b), which was consistent with the literature[46]. The immunofluorescence results were further confirmed by western blot (Supplementary Fig. 15). The impact of co-delivery NP on Bcl pathway was validated by additional IHC staining of tumor tissues (Supplementary Fig. 16). A recent study suggested that the activation of Bcl anti-apoptotic pathway can prevent the release of multiple apoptogenic molecules from mitochondria and rescue PDAC cells from execution of apoptosis, which supported drug resistance in the long run[46,47]. CDK4/6*i*-induced degradation of Mcl-1 elicited vulnerabilities toward Bcl-2 inhibition in PDAC[46]

**Fig. 5 Mechanistic studies revealed co-delivery NP induced regulation of Bcl-2 anti-apoptotic protein, which could be exploited for combination therapy using co-delivery NP plus a Bcl inhibitor. a** Representative immunohistochemical staining of cleaved caspase-3 (CC-3) on tumor sections from the efficacy study in orthotopic PANC-1 model (scale bar: 50 μm). Immunofluorescence staining of CC-3 (green) in co-delivery NP-treated PANC-1 cells confirmed the absence of CC-3 induction (scale bar: 20 μm). Quantification of fluorescence intensity ($n = 6$ ROIs) showed that co-delivery NP significantly enhanced the level of Bcl-x$_L$ ($p = 2.4 \times 10^{-5}$) and reduced the expression of Mcl-1 ($p = 4.4 \times 10^{-4}$) compared to control. **b** Immunofluorescence staining of Bcl-x$_L$ (green), Mcl-1 (red), and Bcl-2 (green) in co-delivery NP-treated PANC-1 cells demonstrate the impact on the expression of Bcl anti-apoptotic family proteins (scale bar: 20 μm). Quantification of fluorescence intensity on confocal images ($n = 6$ ROIs). **c** In vitro anti-proliferative efficacy by BrdU assay ($n = 3$) and CC-3 expression by immunofluorescence staining ($n = 6$ ROIs) in PANC-1 cells. PANC-1 cells were incubated with co-delivery NP (1 μM PAL and 4.3 μM HCQ) plus ABT-737 (1 μM) for days 0–3 and replaced by DMEM for days 4–7. Combined use of co-delivery NP plus ABT-737 significantly reduced cell proliferation ($p = 0.011$) and increased the rate of apoptosis ($p = 9 \times 10^{-6}$) compared to co-delivery NP. Representative images of CC-3 (green) staining were shown (scale bar: 20 μm). **d** Efficacy study with co-delivery NP plus ABT-737 in subcutaneous PANC-1 xenograft model. Mice ($n = 4$) received 5 i.v. injection of co-delivery NP (210 mg particles/kg/injection; PAL: 10 mg/kg/injection; HCQ: 33 mg/kg/injection), each i.v. injection was followed by i.p. injection of ABT-737 at 50 mg/kg. Combined use of co-delivery NP and ABT-737 led to further improvement of antitumor effect compared to co-delivery NP alone ($p = 0.002$). **e** Immunohistochemical staining of Ki67 and CC-3 on tumor sections collected at the conclusive stage (scale bar: 50 μm). Data were obtained from one experiment without repetition. Data represent mean ± SD; statistical difference was evaluated by one-way ANOVA followed by Tukey's post hoc test (*$p < 0.05$). Source data are provided as a Source data file.

and could potentially confer the sensitivity to BH3 mimetic, such as pan-Bcl-2 inhibitor, ABT-737[48].

In addition, CDK4/6i-induced senescence was also investigated, which would be another potential possibility accounting for the lack of apoptosis induction. Different from PAL NP, co-delivery NP did not trigger senescence, evidenced by the analysis of senescence-associated beta-galactosidase and senescence-associated secretory phenotypes (Supplementary Fig. 17). The underlining mechanism is out of the scope of current work and requires further investigation.

Collectively, the co-delivery NP has demonstrated as an effective option in subcutaneous and orthotopic PANC-1 models. In addition to anti-proliferative effect, co-delivery NP led to the regulation of the Bcl-2 family proteins, which could be exploited to further improve the efficacy of co-delivery NP in PDAC.

**Co-targeting Bcl pathway enhances the performance of co-delivery NP.** For proof of principle, we chose ABT-737 to combine with co-delivery NP, because this inhibitor directly binds to multiple Bcl anti-apoptotic proteins to tilt the balance between pro- and anti-apoptotic signaling in favor of cellular apoptosis[49]. Preclinically, ABT-737 can effectively reverse the protective effect associated with the overexpression of Bcl family proteins and has been studied in combination with multiple antitumor therapeutics[50–52] to elicit beneficial efficacies.

We first investigated the combined use of co-delivery NP plus ABT-737 in vitro (Fig. 5c). PANC-1 cells received co-delivery NP (equivalent to 1 μM PAL and 4.3 μM PAL) plus ABT-737 (1 μM) from days 0 to 3. The culture medium was replenished with fresh Dulbecco's modified Eagle medium (DMEM) medium, followed by an additional culture for 4 days (replenishment approach). Controls included saline, ABT-737 alone, and co-delivery NP alone at identical dose. BrdU proliferation assay (Fig. 5c) suggested that the combination of co-delivery NP and ABT-737 improved the anti-proliferative efficiency compared to single agents. More importantly, immunofluorescence on PANC-1 cells treated with co-delivery NP and ABT-737 (Fig. 5c) revealed significantly enhanced CC-3 expression compared to each of the single agents. In addition to the replenishment scheme, we have also studied a sequential approach (co-delivery NP treatment from days 0 to 3; ABT-737 treatment from days 4 to 7) (Supplementary Fig. 18), which exhibited less competent level of cell growth inhibition. In addition to potency, the replenishment scheme has been chosen for the following study because it mimics the on and off treatment regimen in vivo.

Subsequently, we performed an independent efficacy study in which the treatment of co-delivery NP was tested with/without ABT-737 (i.p. 50 mg/kg/injection) in subcutaneous PANC-1 xenograft model (Fig. 5d). We also included another group of mice receiving ABT-737 alone[53]. Co-delivery NP have yielded reproducible anticancer efficacy (similar to Fig. 4a). However, a tendency toward tumor relapse was observed ~20 days post the last i.v. injection. This contrasted with the treatment using co-delivery NP plus ABT-737, which showed further enhanced shrinkage of PANC-1 tumor for up to 50 days with the absence of tumor recurrence. In the groups receiving treatment of saline or ABT-737-alone, endpoints were reached at day 38, in which significantly impaired ambulation resulted from maximum allowable tumor burden.

At the conclusive stage of the experiment, tumor tissues were collected and subjected to IHC staining of Ki67 (proliferative marker) and CC-3 (apoptotic marker). When combined with ABT-737, co-delivery NP led to consistently restrained Ki67 expression at day 50 (Fig. 5e; see low-magnification data in Supplementary Fig. 19), which corroborated the long-lasting tumor inhibition effect in Fig. 5d. Furthermore, the consistent tumor inhibition from co-delivery NP combined with ABT-737 was accompanied by enhanced expression of CC-3 (Fig. 5e and Supplementary Fig. 19), which marked the onset of apoptosis at the tumor site. This differed from the moderate recovery of Ki67 expression in the co-delivery NP-treated group, which is associated with a tendency toward tumor relapse from the last i.v. injection onwards to day 50.

## Discussion

In this study, we demonstrate the use of nano-enabled approach for implementing CDK4/6i combination therapy in PDAC. A ratiometric co-encapsulation of PAL plus an autophagy inhibitor HCQ could lead to a synchronized PK profile and secured drug ratio, which are key factors for optimal drug synergy in vivo. Ratiometric co-delivery of PAL/HCQ generate potent anticancer effect in subcutaneous and orthotopic PDAC models, outperforming a list of controls including free drug mixture. Notably, following the repetitive administrations of co-delivery NP, Bcl anti-apoptosis pathway can be activated. The use of the pan-Bcl-2 inhibitor, ABT-737, further potentiates the effectiveness of the co-delivery NP, leading to a long-lasting anti-PDAC effect for up to 50 days.

CDK4/6i have been tested for treating a list of solid tumor types, with particular success for certain breast cancer. Since the first approval in 2015, the expanded use of CDK4/6i in additional tumor types has been an area of great interest due to the potency of this drug, as well as its unique mode of action. However, ample evidence has suggested that solid tumors (including PDAC) usually

did not harbor a durable response to CDK4/6i monotherapy, and combination therapy could be a practical alternative approach. Although free drug combination has been a promising story in a few preclinical studies, restrictions may occur with respect to independent PK profile, inconsistent drug concentrations, poor intratumoral drug access, and even toxicity. While CDK4/6i impacts the specific component of the cell cycle machinery with significantly reduced toxicity compared to pan-CDK inhibitors, FDA recently published a safety warning for all three approved CDK4/6i on the rare but severe inflammation in the lungs, which may lead to fatality (https://www.fda.gov/drugs/drug-safety-and-availability/fda-warns-about-rare-severe-lung-inflammation-ibrance-kisqali-and-verzenio-breast-cancer). As a potential strategy to address the above challenges, CDK4/6i ratiometric co-delivery platform provides compartmentalized drug distribution in vivo, including the capability to maintain predetermined ratio for optimal drug synergy and toxicity reduction.

Beyond PAL/HCQ combination, versatile CDK4/6i combinations have emerged, such as additional autophagy inhibitors, chemotherapeutic agents, MEK inhibitors, and immunomodulatory drugs. It is promising to identify structurally and pharmacologically suitable drug pair and to design additional nanocarriers for ratiometrically co-packaged CDK4/6i combination therapy. These tasks would require iterative optimization of nanocarriers, with investigations on formulation design and considerations on the protonated status and stability of drug pairs in the confined space of nanocarriers. If a multi-drug regimen is required, such as PAL + HCQ + ABT-737, our preference is to develop a two-in-one NP while introducing the third API through conventional administration route. This is more practical from the perspective of manufacturing process and quality control. While co-delivery NP have been shown to be effective in proof-of-principle PDAC models, it would be necessary to further fine-tune the formulation with respect to additional considerations, including long-term stability, shelf-life, prevention of lipid peroxidation, capability of lyophilization, etc. Moreover, recent studies suggested that CDK4/6 inhibition may lead to augmented antitumor immunity, through activating immune surveillance and modulating tumor microenvironment[40,54]. Further in-depth study is needed to explore the immunological effect of encapsulated CDK4/6i in an immunocompetent model, such as a Rb+ syngeneic mouse model.

## Methods

**Chemicals and reagents.** Tetraethylorthosilicate, triethanolamine, cetyl-trimethylammonium chloride solution (25 wt% in water), triethylamine (TEA), HCQ, Sepharose CL-4B matrix, and crystal violet reagent were purchased from Sigma-Aldrich. DSPC, DSPE-PEG$_{2000}$, and Chol were purchased from Avanti Polar Lipids. Sucrose octasulfate (SOS) sodium salt was obtained from Toronto Research Chemicals, Inc., and converted to sucrose octasulfate triethylamine salt (TEA$_8$SOS) through an ion exchange chromatography process[24,26]. PAL was purchased from LC Laboratories. ABT-737 was purchased from AdooQ Bioscience. Matrigel matrix basement membrane was purchased from BD Bioscience. Premo™ autophagy tandem sensor RFP-GFP-LC3B Kit, Tali cell cycle kit, and CellROX™ Green flow cytometry reagents were purchased from Thermo Fisher Scientific. BrdU cell proliferation assay kit and senescence β-galactosidase staining kit were purchased from Cell Signaling Technology.

**Cell culture.** All human PDAC cell lines were from the American Type Culture Collection (ATCC). PANC-1 cells and MIA PaCa-2 cells were maintained in DMEM supplemented with 10% fetal bovine serum (FBS), 100 U/mL penicillin, 100 μg/mL streptomycin, 2 mM L-glutamine, and 1 mM sodium pyruvate. BxPC-3 cells and AsPC-1 cells were maintained in RPMI 1640 medium supplemented with 10% FBS, 100 U/mL penicillin, and 100 μg/mL streptomycin. HPAF-II cells were maintained in Eagle's Minimum Essential medium supplemented with 10% FBS, 100 U/mL penicillin, and 100 μg/mL streptomycin. To facilitate bioluminescence imaging (BLI) of tumor growth, PANC-1 cells were permanently transfected with a luciferase-lentiviral vector in the UCLA vector core facility[37]. Briefly, $1 \times 10^6$ PANC-1 cells immersed in 1 mL complete DMEM were transduced with luciferase-lentiviral vector in 6-well tissue culture plates. The viral containing media was

removed after 16 h, and the cultures were replenished with fresh DMEM media. Cells were allowed to proliferate to a population size of $5 \times 10^6$. Limiting dilution was used to select individual cell that express the highest luciferase. The highest luciferase expressing clone (refers as PANC-1-luc) out of 10 single clones was used for further experiments.

**In vitro proliferation assay for screening PAL/HCQ ratio.** PDAC cell lines were treated with free PAL/HCQ pair for 72 h at various combination of molar ratios and concentrations. Cells treated with HCQ alone and PAL alone at equivalent concentration were also tested. Cell proliferation was determined by BrdU proliferation assay kit (Cell Signaling Technology) as per the manufacturer's instruction ($n = 3$) and detected on SpectraMax M5e microplate reader with the SpectraMax SoftMax Pro7 software. The absorbance at 450 nm was normalized to saline-treated control to give the percentage of proliferating cells. The results were analyzed with the CompuSyn software to obtain CI and IC$_{50}$ values. Crystal violet staining was also performed to visualize proliferation. Treated PDAC cells were fixed with ice-cold methanol for 10 min and stained with 0.5 wt% crystal violet solution in methanol/water (1:3, v/v) for 20 min.

**Preparation of trapping agent TEA$_8$SOS-loaded LB-MSNP.** Bare MSNPs were synthesized using a sol-gel reaction[30,32]. LB coating was performed with an ethanol exchange method[24]. Briefly, a mixture of DSPC (1.28 g), Chol (0.42 g), and DSPE-PEG$_{2000}$ (0.22 g), equivalent to a molar ratio of 3:2:0.15, was dissolved in 4 mL ethanol at ~65 °C. Forty milliliters of a preheated (~65 °C) aqueous solution, containing 40 mg/mL MSNP and 80 mM TEA$_8$SOS trapping agent, was poured into the lipid solution. The mixture was subjected to probe sonication (52 W) using a 15 s/5 s on/off cycle for 30 min. The un-encapsulated TEA$_8$SOS was removed by Sepharose CL-4B size-exclusion column using a HEPES-buffered dextrose solution (5 mM HEPES, 5 wt% dextrose, pH 6.5) for elusion. TEA$_8$SOS-loaded LB-MSNP were collected and stored at 4 °C for further use.

**Ratiometric remote loading of PAL/HCQ.** We performed iterative optimization for the ratiometric co-encapsulation of PAL/HCQ pair. The loading conditions included the PAL/HCQ/LB-MSNP feed ratios, pH, incubation time, etc. The detailed description on the loading study is in Supplementary Fig. 6. Here we provide the optimized loading procedures for co-delivery NP with a PAL/HCQ feed ratio of 1:5 (molar ratio) as an example. PAL (4.2 mg, 8.7 μmol) and HCQ (18.6 mg, 43 μmol) were dissolved in 10 mL of HEPES-buffered dextrose solution (5 mM HEPES, 5 wt% dextrose at pH 6.5) and mixed well. Ten milliliters of TEA$_8$SOS-loaded LB-MSNP suspension in HEPES-buffered dextrose solution (5 mg particles/mL) was added and incubated at 65 °C for 30 min. The mixture was then quenched on ice for another 30 min. The drug-loaded co-delivery NP were washed 3 times using a HEPES-buffered NaCl solution (4.05 mg/mL HEPES, 8.42 mg/mL NaCl, pH 7.2) by centrifugation at 15,000 rcf. The co-delivery NP were re-suspended in saline and filtered with a 0.45-μm syringe filter, followed by a 0.2-μm filter for sterilization.

**Physicochemical characterization of co-delivery NP.** The co-delivery formulation was characterized for size, charge, morphology, loading capacity, and PAL/HCQ ratio. The morphology including the intactness of LB coating was visualized by cryoEM (TF20 FEI Tecnai-G2) with the DigitalMicrograph GMS3 software. The hydrodynamic size and ζ-potential were measured on ZETAPALS (Brookhaven Instruments Corporation), with the NPs suspended in saline at a particle concentration of 100 μg/mL. To determine the drug-loading capacity, co-delivery NP were treated with acidic methanol solution (0.1 mol/L phosphoric acid/methanol, 1:4 v/v) to remove LB coating and extract PAL and HCQ. Briefly, 100 μL of co-delivery NP (10 mg particles/mL saline) was diluted with 900 μL acidic methanol solution and subjected to bath sonication for 30 min. The solution was then centrifuged at 15,000 rcf for 10 min, and the supernatant was collected. The extraction process was repeated 3 times and drug containing supernatants were filtrated through 0.22-μm filters for HPLC analysis on PerkinElmer Altus A-10 HPLC System with Brownlee SPP 2.7 μm C18 column, followed by UV detection at 340 nm. The mobile phase contained 3% triethylammonium acetate aqueous solution (pH 5.6) and acetonitrile (73:27, v/v) at a flow rate of 0.2 mL/min. The typical retention time was identified as 2.4 min for HCQ and 5.9 min for PAL, respectively. The representative HPLC result is shown in Supplementary Fig. 10.

**In vitro anti-proliferation study of co-delivery NP.** PANC-1 cells treated with co-delivery NP at various loading ratios were also tested by BrdU proliferation assay. A series of PAL concentrations (0–20 μM) and HCQ concentrations (0–400 μM) were tested. Cells treated with HCQ NP and PAL NP at equivalent concentration were also tested. Cell proliferation was determined by BrdU proliferation assay kit (Cell Signaling Technology) as per the manufacturer's instructions. The absorbance at 450 nm were normalized to saline control to give the percentage of proliferating cells. The results were analyzed with the CompuSyn software to obtain CI and IC$_{50}$. To probe cell cycle, PANC-1 cells were treated with co-delivery NP (1:4.3 loading molar ratio, PAL = 2.5 μM, HCQ = 10.8 μM) for 72 h ($n = 3$). Cells were fixed with 4% paraformaldehyde for 10 min at room temperature. Then the cells were washed with phosphate-buffered saline (PBS) and

incubated with Tali cell cycle solution (Thermo Fisher) that contains propidium iodide, RNase A, and Triton X-100 for 30 min in dark. To obtain single-cell suspension, the cells were passed through a 70-μm cell strainer. Cells were detected on a flowcytometer (LSRII, Becton Dickinson) with the BDFACSDiva™ software 7.0. The cell cycle analysis was performed with the FlowJo software (v7.6.1).

**In vitro autophagy analysis and ROS detection**. To characterize autophagy and ROS production, PANC-1 cells were treated with co-delivery NP, HCQ NP, or PAL NP for 72 h, at equivalent drug concentration (PAL = 2.5 μM, HCQ = 10.8 μM). For TEM characterization of autophagic vacuoles, PANC-1 cells treated by drug-laden NPs were fixed with 2% glutaraldehyde solution (EM grade) overnight at 4 °C. Further sample preparation and sectioning were performed by BRI Microscopic Techniques and Electron Microscopy Core Facility at UCLA. Briefly, cell samples were then post-fixed in osmium tetraoxide, dehydrated in ethanol, and embedded in resin. Ultra-thin sections (~80 nm) were cut and placed on copper grids. The sections were stained with uranyl acetate and lead citrate, followed by observation by TEM (Biotwin, JEOL). For the detection of ROS production, PANC-1 cells treated by drug-laden NPs were stained by CellROX™ Green flow cytometry kit as per the manufacturer's instructions for 30 min at 37 °C. Cells were washed with PBS and harvested. Cells were passed through a 70-μm cell strainer and then subjected to flow cytometry for ROS detection. For the GFP-RFP-LC3 dual reporter assay, PANC-1 cells were pre-transfected with the Premo™ autophagy tandem sensor RFP-GFP-LC3B Kit (Thermo Fisher Scientific) before incubated with drug-laden NPs for 72 h. The cells were then counter-stained with Hoechst 33342 and imaged by confocal microscopy (SP8-SMD, Leica) with the LAS X software.

**Immunofluorescence staining**. PDAC cells were treated with co-delivery NP, HCQ NP, or PAL NP for 72 h at equivalent drug concentration (PAL = 2.5 μM, HCQ = 10.8 μM). PANC-1 cells were then fixed with 4% paraformaldehyde, permeabilized with 0.2% Triton X-100, and blocked with 3 wt% bovine serum albumin (BSA). Primary antibodies included anti-phospho-Rb (1:1000, cat#8516, Cell Signaling Technology), anti-p62 (1:100, cat#ab91526, Abcam), anti-Ki67 (1:500, cat#ab15580, Abcam), anti-CC-3 (1:400, cat#9664, Cell Signaling Technology), anti-Bcl-2 (1:150, cat#ab182858, Abcam), anti-Mcl-1 (1:500, cat#ab32087, Abcam), or anti-Bcl-$x_L$ (1:500, cat#ab32370, Abcam) accordingly. Cells were further incubated with secondary antibodies, including Alexa Fluor 594 goat anti-rabbit immunoglobulin G (IgG; 1:500, cat#A11012, Thermo Fisher Scientific) or Alexa Fluor 488 goat anti-rabbit IgG (1:500, cat#A11008, Thermo Fisher Scientific), and counterstained with 4,6-diamidino-2-phenylindole. The immunofluorescence staining was visualized by confocal microscopy. Fluorescence intensity of biomarkers including Ki67, Bcl-$x_L$, Mcl-1, and Bcl-2 was quantified with the ImageJ software (v1.5.2) and normalized to the number of cells as indicated by nuclei stain.

**Western blot analysis**. The expression of Rb, pRb, Ki67, p62, LC3B, Bcl-$x_L$, and Mcl-1 was determined by western blot analysis. PANC-1 cells were treated with free co-delivery NP, HCQ NP, or PAL NP for 72 h at equivalent drug concentration (PAL = 2.5 μM, HCQ = 10.8 μM). Protein was extracted in RIPA buffer supplemented with proteinase and phosphatase inhibitor cocktail (Cell Signaling Technology), followed by electrophoresis on Novex™ tris-glycine mini gels (Invitrogen). The proteins were subsequently transferred to a polyvinylidene difluoride membrane. After blocking in 5% BSA (for pRb, p62, and vinculin) or 5% non-fat dry milk (for Rb, Ki67, Bcl-$x_L$ Mcl-1, LC3B, and β-actin), the membrane was sequentially incubated with primary antibodies, including anti-Rb (1:1000, cat#9313, Cell Signaling Technology), anti-phospho-Rb (1:1000, cat#8516, Cell Signaling Technology), β-actin (1:1000, cat#3700, Cell Signaling Technology), anti-SQSTM1/p62(1:1000, cat#ab56416, Abcam), anti-Bcl-$x_L$(1:1000, cat#ab32370, Abcam), anti-Mcl-1 (1:1000, cat#ab32087, Abcam), anti-Vinculin (1:1000, cat#13901, Cell Signaling Technology), anti-Ki67 (1:5000, cat# ab92742, Abcam), and anti-LC3B (1:1000, cat#2775, Cell Signaling Technology). Horseradish peroxidase (HRP)-conjugated secondary antibodies included anti-rabbit HRP-linked IgG (1/1000, cat#7074, Cell Signaling Technology) and anti-mouse HRP-linked IgG (1/1000, cat#7076, Cell Signaling Technology). The blots were developed by ECL solution. The intensity of blots was quantified by the Image J software (v1.5.2).

**Establishment of subcutaneous PANC-1 pancreatic tumor model**. Female homozygous athymic Balb/c nude mice (6–8 weeks) were purchased from Charles River Laboratories and maintained in pathogen-free condition. The protocol, which is approved by Animal Research Committee, includes standard operating procedures for animal housing (filter-topped cages; room temperature at 23 ± 2 °C, 60% relative humidity; 12 h/12 h light/dark cycle) and hygiene status (autoclaved food and acidified water). To establish a subcutaneous PANC-1 xenograft model, $8 \times 10^6$ PANC-1 cells in 100 μL of DMEM/Matrigel mixture (1:1, v/v) were subcutaneously injected to the lower right flank of mice. The efficacy study was performed in the tumor-bearing mice when subcutaneous tumors had grown to ~5 mm. For the PK and intratumoral biodistribution studies, the tumor-bearing mice were used when subcutaneous tumors had grown to a size of ~10 mm.

**Establishment of orthotopic PANC-1 pancreatic tumor model**. Orthotopic PANC-1 xenograft model was established according to our prior publication[24]. Briefly, the animals were anesthetized with isoflurane, and the surgical site was sterilized by repetitive application of betadine and 70% ethanol. Animals were placed in an appropriate position for surgery on a water heating pad in the tissue culture hood, and the surgical site was draped with sterile gauze. A surgical incision of 0.5–0.7 cm was made in the left flank to expose the injection site, following with an injection of 50 μL of DMEM/Matrigel (1:1, v/v) containing $8 \times 10^6$ PANC-1-luc cells into the tail of the pancreas by a 27-gauge needle. The fascial layers were then closed with absorbable sutures (PDS II, Ethicon) and the skin with non-absorbable sutures (PROLENE, Ethicon). The mice were kept on the warming pads until full recovery from the anesthesia and then transferred to clean cages and maintained in pathogen-free condition. The efficacy study was performed in the tumor-bearing mice 4 weeks post inoculation.

**In vivo PK and biodistribution study**. Subcutaneous PANC-1 tumor-bearing mice received single i.v. injection of co-delivery NP (with the particle dose of 210 mg/kg, dispersed in 200 μL saline) or free PAL/HCQ pair, with an equivalent PAL dose of 10 mg/kg and HCQ dose of 33 mg/kg that correlated to a PAL/HCQ molar ratio of 1:4.3 (n = 3). Blood samples were collected at 5 min, 3, 6, 24, and 48 h. After separation of the plasma fraction, PAL/HCQ was extracted in an acidic methanol solution (0.1 mol/L phosphoric acid/methanol, 1:4 v/v). The drug concentration was measured by UPLC-MS (Waters LCT Premier ESI) with the MassLynx 4.1 software, using gradient elution of 0.3% formic acid in water and 0.3% formic acid in acetonitrile, at a flow rate of 0.2 mL/min. The ratio of PAL/HCQ was calculated. The $t_{1/2}$ was determined via the PKSolver software. Comparison was made between co-delivery NP and free PAL/HCQ pair.

**Intratumoral drug content and biodistribution**. Subcutaneous PANC-1 tumor-bearing mice received single i.v. injection of co-delivery NP or free PAL/HCQ pair, with an equivalent PAL dose of 10 mg/kg and an HCQ dose of 33 mg/kg (n = 3). The volume of injection was 200 μL for each animal. Mice were sacrificed 24 and 48 h post injection to collect tumor tissues. Tumors were weighed and homogenized, and PAL/HCQ was extracted in acidic methanol solution. The drug concentration was determined by UPLC-MS as described in PK study. A confirmative study was also performed in orthotopic PANC-1 xenograft model, in which mice received i.v. injection of DyLight680-labeled co-delivery NP[24]. Mice were sacrificed 48 h post injection; the tumor tissue and major organs were collected for ex vivo IVIS imaging to detect the NIR fluorescence and overlaid with the bioluminescence signal from the tumor tissue. The tumor tissues were also cryo-embedded in OCT reagent and cryo-sectioned for immunofluorescence staining. Blood vessels were stained with a primary anti-CD31 antibody (1:500, cat#53708, BD Bioscience), followed by an Alexa Fluor 488-conjugated secondary antibody (1:500, cat#A11006, Thermo Fisher Scientific). Tumor sections were visualized by confocal microscopy.

**Efficacy study in subcutaneous PANC-1 model**. Subcutaneous PANC-1 tumor-bearing mice were randomly assigned to 7 groups (n = 4)[55–57]. Mice received i.v. injection of (1) saline, (2) free PAL/HCQ, (3) PAL-only NP, (4) HCQ-only NP, (5) free PAL + HCQ-only NP, (6) free HCQ + PAL-only NP, and (7) co-delivery NP, with an equivalent PAL dose of 10 mg/kg/injection and a HCQ dose of 33 mg/kg/injection. Five i.v. injections were administered as shown in Fig. 4a. The volume of injection was 200 μL/injection for each animal. Tumor size was monitored by measuring the length and width with an electronic caliper and was calculated according to the formula: tumor size = (length × width$^2$)/2[29]. Mice were sacrificed on day 21 from the first injection, and tumor tissues were collected to compare the antitumor efficacy.

**Confirmative efficacy study in the orthotopic model**. Orthotopic PANC-1 tumor-bearing mice were randomly assigned to 5 groups (n = 4). Mice received i.v. injection of saline, free PAL/HCQ, HCQ NP, PAL NP, and co-delivery NP, with an equivalent PAL dose of 10 mg/kg/injection and a HCQ dose of 33 mg/kg/injection. Five i.v. injections were administered and the scheme for injection was the same as the efficacy study in subcutaneous models. To monitor the development of orthotopic tumor, mice received i.p. injection of D-luciferin (75 mg/kg) and subjected to live BLI with IVIS system (Xenogen). Tumor burden was quantitatively expressed as BLI intensity in the operator-defined ROI[24,26,37]. The bioluminescence intensity in each ROI was quantified by the IVIS Living Image software v4.5 and plotted versus time. Mice were sacrificed on day 30 from the first injection. Tumor tissues were collected for IHC analysis of multiple biomarkers.

**Assessment of the potential toxicity of co-delivery NP**. Blood samples and major organs were harvested from the efficacy study in orthotopic models. Blood samples were subjected to complete blood count and biochemistry analysis. Major organs were fixed overnight in 10% formalin, followed by paraffin embedding and sectioning. Tissue sections of 4-μm thickness were mounted on glass slides and stained with hematoxylin and eosin.

**Efficacy study of co-delivery NP in combination with ABT-737.** Subcutaneous PANC-1 tumor-bearing mice were randomly assigned to 4 groups, including saline, free ABT-737, co-delivery NP, and co-delivery NP + ABT-737 ($n = 4$). Co-delivery NP were i.v. injected at PAL dose of 10 mg/kg/injection and an HCQ dose of 33 mg/kg/injection. For the group receiving co-delivery NP + ABT-737 combination treatment, each i.v. injection was followed by i.p. injection of ABT-737 (50 mg/kg/injection). Mice receiving i.p. injection of ABT-737 alone or saline control were also tested. A total of 5 i.v. injections were administered as shown in Fig. 5d. Tumor size was monitored and plotted versus time. At the conclusive stage, tumor tissues were collected for IHC analysis of multiple biomarkers.

**IHC staining.** PDAC tumor tissues were fixed in 10% formalin solution overnight, followed by paraffin embedding and sectioning. Briefly, tumor sections of 4-μm thickness were de-paraffinized, incubated in 3% methanol–hydrogen peroxide, followed by incubation with 10 mM EDTA at 95 °C using the Decloaking Chamber (Biocare Medical, DC2012). The slides were incubated with individual primary antibodies for 1 h including anti-phospho-Rb (1/200, cat#8516, Cell Signaling Technology), anti-Ki67 (1/100, cat#ab15580, Abcam), anti-CC-3 (1/200, cat#9664, Cell Signaling Technology), anti-Mcl-1 (1/200, cat#ab32087, Abcam), or anti-Bcl-$x_L$ (1/400, cat#ab32370, Abcam). After washing, the slides were further incubated with HRP-conjugated secondary antibodies at room temperature for 30 min. After rinsing with PBST, the slides were incubated with 3,3′-diaminobenzidine and counterstained with hematoxylin. The slides were scanned by an Aperio AT Turbo Digital Pathology Scanner (Leica Biosystems). Quantitative evaluation of antigen expression was performed with the Aperio ImageScope software 12.3.

**Statistics.** Comparative analysis of differences between groups was performed using one-way analysis of variance followed by Tukey's post hoc test (Origin 2018, OriginLab). Values were expressed as mean ± SD. For all statistical analyses, $p < 0.05$ was considered statistically significant.

**Reporting summary.** Further information on research design is available in the Nature Research Reporting Summary linked to this article.

## Data availability

The source data underlying Figs. 1a, c, e, 2c, f, g, 3a, b, 4a, d, and 5b–d and Supplementary Figs. 3, 4, 7, 12–14, 16–19 and all the raw western blots corresponding to the SDS-PAGE gels (Fig. 1h and Supplementary Figs. 1, 9, and 15) are provided as a Source data file. All the other data supporting the findings of this study are available within the article and its supplementary information files and from the corresponding author upon reasonable request. A reporting summary for this article is available as a Supplementary Information file. Source data are provided with this paper.

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

## Acknowledgements
This study was supported by the U.S. Public Health Service Grant, 1U01CA198846. This study was partially supported by a research grant, DISC2-08824, funded by California Institute for Regenerative Medicine. We acknowledge use of the IVIS imaging facilities in the Preclinical Imaging Technology Center, the Translational Pathology Core Laboratory for histology H&E and IHC staining, the Electron Imaging Center for Nanomachines for CryoEM, the Molecular Instrumentation Center for UPLC-MS, and the CNSI Advanced Light Microscopy/Spectroscopy Shared Facility for confocal fluorescent microscopy at UCLA.

## Author contributions
H.M. designed and supervised the research project. Y.J. and X.L. designed and performed the studies, analyzed the results, and wrote the manuscript. Y.J., X.L., J.L., X.X., M.H., J.J., and Y.-P.L. performed the experiments. T.D. provided helpful suggestions to the work.

## Competing interests
The authors declare the following competing interest(s): H.M. is co-founder, board member, and equity holder in Westwood Bioscience Inc. H.M. is also co-founder and equity holder in NAMMI therapeutics. The remaining authors declare no competing interests.
