## [Peer Review File · Nature Communications]

Reviewers' comments:

Reviewer #1 (Expertise: NPs, drug delivery, Remarks to the Author):

In this manuscript, the authors used a liposomal mesoporous silica nanocarrier for co-loading of a CDK4/6 inhibitor (PAL) and an autophagy inhibitor (HCQ) for pancreatic cancer therapy. There are several highlights in this manuscript as following:

1. PAL and HCQ are highly synergistic in inhibiting pancreatic cancer cell proliferation at certain ratios. By utilizing an optimized remote loading method, PAL and HCQ can be ratiometrically loaded into the liposomal mesoporous silica nanocarriers.
2. In vivo pharmacokinetics and intratumoral drug concentration tests confirmed that these two drugs were kept at designed ratios in the blood and the tumor tissue.
3. The nanocarrier-enabled ratiometric drug delivery resulted in superior tumor inhibition effect over other forms in murine pancreatic cancer models.

The authors show strong evidence that the ratiometrically designed nanocarrier will greatly enhance the therapeutic effect of PAL and HCQ over free combinations. This is quite meaningful since the first ratiometric liposome (Vyxeos®) for co-delivery of daunorubicin and cytarabine has been approved by the FDA in 2017.

The manuscript is well organized and well written. In each part, the descriptions are extensively supported by detailed data.

One concern for this manuscript is the payload chosen in utilizing this nanocarrier for cancer therapy. Besides biological aspects, chemical properties may be another reason for co-loading PAL and HCQ into the nanocarrier with the described remote-loading method. It will be more meaningful to give a comprehensive description on the ratiometric drug loading capacity of this nanocarrier in other drugs if more drugs are tested in the drug loading experiments.

Reviewer #2 (Expertise: PDAC, autophagy, therapy, Remarks to the Author):

Marginal success with CDK4/6 inhibitors in pre-clinical studies has prompted to develop combination therapies with better drug delivery methods to treat solid tumors. In this manuscript, Ji et al have used combination of CDK4/6 inhibitor Palbociclib (PAL) with autophagy inhibitor Hydroxychloroquine (HCQ) in a ratiometrically designed mesoporous silica nanoformulation (nanoparticle or NP) and showed that this method of drug delivery is more potent in part due to improved stability and bio-availability of the drugs in tumors.

Although this nanoparticle-based formulation shows better efficacy as compared to free PAL and/or HCQ, they observed tumor recurrence because of the expression of BCL-xl anti-apoptotic machinery. Combining a pan BCL2 inhibitor (ABT-737) with NP(PAL+HCQ); the authors could improve the antitumor effect of NP and suggested a rationale to use this three-drug combination for pancreatic cancer therapy. While this article is important and provides evidence of an efficacious drug delivery method, the concept of using PAL+HCQ is has been shown in other solid tumor studies (for example Nature Communications volume 8, Article number: 15916 (2017)). Use of limited number of cell lines and missing controls in their experiments are major concerns of the paper. Immune cells have profound roles in the growth of KRAS driven pancreatic cancer (PDAC). As CDK4/6i(+MEKi) mediated growth inhibition of KRAS mutant tumors involves of immune cells, (Science. 2018 Dec 21;362(6421):1416-1422. doi: 10.1126/science.aas9090), the formulation should be tested in immune-competent animal models. Following suggestions/ concerns need to be addressed to improve the manuscript and novelty of the study.

- 1) The majority of the experiments have been done in only one cell line (PANC-1). As PDAC cells are very heterogeneous, the in-vitro study should be conducted in at least 5-6 cancer cell lines.
- 2) The experiments should be performed in orthogonal methods to substantiate the conclusions. Immunoblots should be provided for Fig 1B, 2J, 5B and 5D. Please show p62 in immunoblot as well.
- 3) Fig 4C. : As presented; the NPs accumulate in near CD31+ tumor blood vessels. How does the drug reach in the CD31 negative areas? Do those areas maintain proliferation? A co-staining with Ki67 would be more informative.
- 4) The histology sections are presented only in higher magnification. As this can be subjective, the authors should provide lower magnification views for a better assessment. A proper quantification is missing and should accompany every histological analysis.
- 5) Many studies have shown CDK4/6 inhibition alone or combined with other drugs invoke a senescence response. Senescence is associated with the upregulation of BCL2 family of proteins. In this study, the authors found BCL-xl is upregulated; however, senescence is not associated with their drug treatment. This is an important observation that needs to be substantiated with proper controls. A) In Fig S10, a proper positive control for SA-B-gal staining should be presented to show that the assay is working in their setting. B) As discussed before this phenomenon should be proved in multiple cell lines. C) The authors can perform quantitative RT-PCR to show that senescence-associated secretory phenotype (SASP) related transcripts are not regulated in their assays.
- 6) As discussed, PDAC has a unique fibrous micro-environment. Not only it is impermeable to drug formulations, but also the cross-talk of immune cell and tumor cell provide a challenge to treat PDAC. As authors claim a more potent drug delivery method, the efficacy of NP or NP+ABT737

should be performed in immunocompetent PDAC models (i.e. Kras-p53 GEMMs) or syngeneic orthotopic transplantation models.

Response Letter

Re: Manuscript NCOMMS-19-2752

We thank the reviewers for thorough review of our manuscript, and making suggestions to improve the quality of the paper. For ease of the review, we prepare our responses in a point-to-point fashion.

Reviewer #1

Comment 1: In this manuscript, the authors used a liposomal mesoporous silica nanocarrier for co-loading of a CDK4/6 inhibitor (PAL) and an autophagy inhibitor (HCQ) for pancreatic cancer therapy. There are several highlights in this manuscript as following:

1. PAL and HCQ are highly synergistic in inhibiting pancreatic cancer cell proliferation at certain ratios. By utilizing an optimized remote loading method, PAL and HCQ can be ratiometrically loaded into the liposomal mesoporous silica nanocarriers.
2. In vivo pharmacokinetics and intratumoral drug concentration tests confirmed that these two drugs were kept at designed ratios in the blood and the tumor tissue.
3. The nanocarrier-enabled ratiometric drug delivery resulted in superior tumor inhibition effect over other forms in murine pancreatic cancer models.

Response: Thank you. We are glad that the reviewer was convinced and stated that PAL and HCQ combo is “highly synergistic in inhibiting pancreatic cancer cell proliferation at certain ratios”. We also appreciate the reviewer’s positive comments on our PK and biodistribution data. We are glad that the reviewer was convinced that our particle led to “superior tumor inhibition effect” over controls.

Comment 2: The authors show strong evidence that the ratiometrically designed nanocarrier will greatly enhance the therapeutic effect of PAL and HCQ over free combinations. This is quite meaningful since the first ratiometric liposome (Vyxeos[®]) for co-delivery of daunorubicin and cytarabine has been approved by the FDA in 2017.

Response: Thank you. We agree with the reviewer that the success of the 1st ratiometric liposome (Vyxeos[®]) is very encouraging, as we mentioned in the introduction section of the original submission. We also appreciate the comment, *i.e.* “this is quite meaningful [investigation]”.

Comment 3: The manuscript is well organized and well written. In each part, the descriptions are extensively supported by detailed data. One concern for this manuscript is the payload chosen in utilizing this nanocarrier for cancer therapy. Besides biological aspects, chemical properties may be another reason for co-loading PAL and HCQ into the nanocarrier with the described remote-loading method. It will be more meaningful to give a comprehensive description on the ratiometric drug loading capacity of this nanocarrier in other drugs if more drugs are tested in the drug loading experiments.

Response: We thank the reviewer’s laudatory comments, *i.e.* “well organized and well written” and “the descriptions are extensively supported by detailed data”.

We agree that chemical property of the paired APIs is an important design consideration in our formulation. To construct our co-delivery nanocarrier, it involves two important requirements, namely 1) use of weak-base API that can be actively imported into the lipid coated MSNP through

the generation of pH gradient, and 2) ensure that the drug pair exhibits similar pKa and solubility, allowing the easy use of single trapping agent for drug co-import.

For each individual payload, the selected weak-base molecule for remote loading should exhibit the following chemical properties: **(i)** organic molecules that include primary, secondary or tertiary amine(s); **(ii)** pKa <11 to allow protonation and entrapment behind the lipid coat; **(iii)** water solubility ranging from around 5 to 25 mg/mL and amphipathic characteristics that allow diffusion across the LB; **(iv)** an octanol/water partition coefficient or log P value of -3.0 to 3.0; **(v)** suitable molecular weight with a geometric size less than MSNP pore size (2-8 nm) to allow entry into the MSNP pores (Meng, *ACS Nano* **2016**, *10*, 2702). We also want to point out that co-package of 2 weak base drugs is more complex than single drug encapsulation in our nanocarrier. For ease of co-package, the selected payload pair should ideally exhibit comparable solubility and pKa, allowing the use of the same trapping agent. Moreover, for drug co-import, the selected loading conditions, such as pH and buffer solutions, should be compatible for both payload molecules. Therefore, the formulation process, based on our experiences, frequently requires a process with multi-factor and iterative optimization, such as type/concentration of trapping agent, incubation time, and particle/HCQ/PAL ratios, etc., which were provided in our original submission (now supplementary Figure S6). For ease of understanding, below scheme summarizes our theory and now appeared as supplementary Figure S5 in the revised manuscript.

Considerations of weak base remote loading

- (i) Organic molecules that include primary, secondary or tertiary amine(s);
- (ii) pKa <11 to allow protonation and entrapment behind the lipid coat;
- (iii) Water solubility ranging from around 5~25 mg/mL and amphipathic characteristics that allow diffusion across the lipid coat;
- (iv) Octanol/water partition coefficient or log P value of -3.0 to 3.0;
- (v) Suitable molecular weight with a geometric size less than MSNP pore size (2-8 nm) to allow entry into the MSNP pores.

Considerations of ratiometric co-import

- (vi) Drug A and Drug B should have similar pKa and solubility values that can be imported into particle at comparable efficiency
- (vii) For ease of drug co-import, use of a single trapping agent is preferred.

Moreover, for review purpose-only, we have performed a pilot study for ratiometric co-package of new CDK4/6 inhibitors (ribociclib, RIB or abemaciclib, ABE) plus HCQ. Similar to PAL, RIB and ABE both are weak-bases with pKa of 8.6 and 7.7, respectively, and their solubility values are of 2.0 mg/mL to 50 mg/mL (pH-dependent). The result is promising; we were able to make RIB/HCQ or ABE/HCQ laden nanoparticles at drug ratios ranging from 5:1 to 1:5.

Reviewer #2

Comment 1: Marginal success with CDK4/6 inhibitors in pre-clinical studies has prompted to develop combination therapies with better drug delivery methods to treat solid tumors. In this manuscript, Ji et al have used combination of CDK4/6 inhibitor Palbociclib (PAL) with autophagy inhibitor Hydroxychloroquine (HCQ) in a ratiometrically designed mesoporous silica

nanoformulation (nanoparticle or NP) and showed that this method of drug delivery is more potent in part due to improved stability and bio-availability of the drugs in tumors.

Response: Thanks for the diligent review. We are glad that the reviewer thinks that the ratiometric co-delivery NP is more potent due to the improved stability and bio-availability at PDAC tumor site.

Comment 2: Although this nanoparticle-based formulation shows better efficacy as compared to free PAL and/or HCQ, they observed tumor recurrence because of the expression of Bcl-x_L anti-apoptotic machinery. Combining a pan Bcl-2 inhibitor (ABT-737) with NP (PAL+HCQ); the authors could improve the antitumor effect of NP and suggested a rationale to use this three-drug combination for pancreatic cancer therapy. While this article is important and provides evidence of an efficacious drug delivery method, the concept of using PAL+HCQ is has been shown in other solid tumor studies (for example Nature Communications volume 8, Article number: 15916 (2017)). Use of limited number of cell lines and missing controls in their experiments are major concerns of the paper. Immune cells have profound roles in the growth of KRAS driven pancreatic cancer (PDAC). As CDK4/6i(+MEKi) mediated growth inhibition of KRAS mutant tumors involves of immune cells, (Science. 2018 Dec 21;362(6421):1416-1422. doi: 10.1126/science.aas9090), the formulation should be tested in immune-competent animal models. Following suggestions/concerns need to be addressed to improve the manuscript and novelty of the study.

Response: There are multiple points in the reviewer’s comment. We addressed it using a bullet point fashion.

(a) Critique on “limited number of cell lines”:

Response: Reviewer #2’s major concern is to use limited number of cell lines. While we have tested our particles in two cell lines (i.e. PANC-1 and MIA PaCa-2) followed by 5 animal experiments (1 biodistribution study, 1 IVIS imaging study and 3 efficacy studies) using subQ and orthotopic PDAC models, we agree that expanding our study using additional cell lines can further improve the comprehensiveness of the paper. Accordingly, we have provided new data using additional 3 PDAC cell lines, i.e. BxPC-3, HPAF-II and AsPC-1. The new findings have been demonstrated in the below picture. Based on the combination index analysis, our data showed that PAL/HCQ exhibited strong synergy at drug ratio of 1:5 in all five PDAC cell lines.

CI	Free PAL/HCQ ratio				
	10/1	2/1	1/1	1/5	1/10
PANC-1	1.19	0.55	0.27	0.23	0.25
MIA PaCa-2	1.47	1.02	0.47	0.27	0.22
BxPC-3	0.95	2.16	1.09	0.32	0.62
HPAF-II	0.84	1.04	0.70	0.48	0.52
AsPC-1	3.50	2.96	0.41	0.32	0.67

↓
strong synergy in vitro

Legend: Use of CompuSyn computer software to determine critical drug ratio for PAL/HCQ combination in 5 PDAC cell lines. This software was used to calculate a synergistic combination index (CI). We define strong synergy when the CI is less than 0.7 (green color). PAL/HCQ showed strong synergy at drug ratio at 1:5 in all five PDAC cell lines. BxPC-3, HPAF-II and AsPC-1 data are new, which have been included in the revised manuscript (Figure 1A). The additional drug synergy study is provided in the revised manuscript (see Figure S2).

Response: PDAC is an almost incurable cancer type. Ample evidence suggest that PDAC is resistant (or transiently responsive) to the existing therapy. This general impression holds true in the case of CDK4/6i therapy (Trends in cancer **2017**, 3, 39). While the ratiometric co-delivery is clearly advantageous than monotherapy or free drug mix, it is not surprising to see tumor recurrence (*i.e.* 30 days post IV injection in mice), presumably due to the tumor aggressiveness and/or signaling redundancy. Our investigation moves additional step by demonstrating the long-lasting anti-PDAC effect using Bcl-2 inhibitor plus our nanocarrier. In our opinion, this discovery is impactful by demonstrating how mechanistic discovery at the nano/bio interface can assist the design and rational use of novel nanocarriers to treat PDAC.

(c) Critique on “using PAL+HCQ has been shown in other solid tumor studies”:

Response: While it is PAL+HCQ has been tested in certain solid tumors (e.g. breast cancer), to our best knowledge, a ratiometric PAL/HCQ nano-formulation was not established and tested in PDAC. In the past ~15 years, there are only 2 FDA-approved new drugs for PDAC. Both of them are nano-enabled, *i.e.* Abraxane® (albumin/paclitaxel nanocomplex) and ONIVYDE® (irinotecan liposome). These early chemo-carriers are not designed to address *in vivo* drug synergy, synchronized PK, and drug ratio. From the drug delivery aspect, this study provides a substantive departure from the current status quo, by introducing a co-delivery nanocarrier for a novel combination (HCQ/PAL) in a single ratiometric-designed carrier in subcutaneous and orthotopic models with secured synergy *in vivo*. Equally important, targeting cell cycle pathway using encapsulated CDK4/6i and addition of a Bcl-2 inhibitor for long-lasting anti-PDAC effect are novel from the perspective of cancer biology.

(d) Critique on not study immunological effect and recommended use of Kras PDAC model in immune competent mice:

Response: Thanks for the comment. While we do agree with Reviewer #2 that CDK4/6i's immune response is of great interest, this is a major topic, which requires independent study from the perspective of experimental design. In terms of Kras PDAC model, our lab frequently used KPC-derived models in our drug delivery studies (ACS Nano **2016**, 10, 2, 2702; J Clin Invest. **2017**, 127, 2007; ACS Nano **2019**, 13, 1, 38). The KPC cells were derived from a spontaneous PDAC tumor from a transgenic *Kras^{LSL-G12D/+}Trp53^{LSL-R172H/+}Pdx1-Cre* mouse (ACS Nano **2016**, 10, 2, 2702). While the KPC-derived model is a great model in terms of Kras mutation, presence of thick stroma, immunocompetence, etc., this model has limitation, precluding the use as a primary discovery model for CDK4/6 inhibitor study. The literature (Gut. 2018, 67, 2142) and our new data (see the inserted picture) showed very low Rb/pRb status in KPC cells, which is the basis of using CDK4/6 inhibitor (labelled in **blue**). This differs from other PDAC lines, which are Rb⁺ (highlighted in **red**). An important clarification is different from KPC pancreatic cancer, Rb protein expression was evidenced in the KP lung cancer (Science 2018, 362, 1416; cited by the reviewer). Therefore, in the revised manuscript, five Rb⁺ PDAC lines were used in this investigation.

At this stage, we envisage that it would be more productive to study immunology of CDK4/6 inhibitor in syngeneic cancer models with Rb/pRb expression. Due to the complexity of cancer immunology, it requires an independent investigation for a nano encapsulated CDK4/6 inhibitor. Moreover, the planning process should also include consideration from the autophagy aspect, such as animal model and p53 status in the tumor (WT vs mutation), which may lead to contrast outcome (J Exp Clin Cancer Resv. 38; 2019). To accommodate the reviewer's suggestion, we have added a discussion section to cover the immunology aspect of our nanocarrier. It reads:

[page 14]. “The availability of a CDK4/6i co-delivery platform would allow us to seek new opportunities to design many other CDK4/6i drug combinations, such as additional

autophagy inhibitors, chemotherapeutic agents, MEK inhibitors, and immunomodulative drugs, which are structurally and biological suitable for co-package. Recently, CDK4/6 inhibition has demonstrated augmented anti-tumor immunity, by activating immune surveillance and modulating tumor microenvironment. It would be very interesting in the future to explore the effect of encapsulated CDK4/6i in a Rb⁺ immunocompetent model. Moreover, if a heterogeneous tumor microenvironment becomes a concern, it would be helpful to contemplate the use of nano-enabled stroma engineering to provide an impact on blood vessel permeability and patency, inhibit drug inactivating enzymes and/or target specific biological factors that play a role in the heterogeneous tumor microenvironment in multiple tumor types.”

Legend: (Left) Scheme illustrating the role of CDK4/6i and Rb protein in the regulation of cancer cell cycle. CDK4/6 inhibitors downregulate phosphorylated Rb (pRb) and prevent the dissociation between Rb and E2F transcription factors. The absence of free E2F transcription factors leads to interrupted G1 to S phase transition and results in G1 cell cycle arrest. Due to the essential role of Rb/pRb, preserved expression of the Rb protein (among others) is being investigated to predict the responsiveness of CDK4/6i. (Right) A list of PDAC cell types were assayed for Rb and pRb via immunoblotting. Different from most PDAC cell lines that are Rb/pRb⁺, KPC cells exhibit non-detectable Rb/pRb level, which is consistent to the literature (Gut. 2018, 67, 2142).

Comment 3: The majority of the experiments have been done in only one cell line (PANC-1). As PDAC cells are very heterogeneous, the in-vitro study should be conducted in at least 5-6 cancer cell lines.

Response: This comment has been largely addressed in response 2(a). Briefly, in addition to PANC-1 cells, our original submission indeed included *in vitro* data in MIA PaCa-2 cells (now supplementary Figure S4), *i.e.* crystal violet staining (panel A), Ki67 proliferation assay (panel B), pRb immunofluorescence staining (panel C), autophagosome staining (panel D), and CellROX staining for reactive oxygen species (panel E). In the revised manuscript, new data using additional 3 cell lines are included, as we discussed above. All the *in vitro* data confirmed what we found in PANC-1 cells.

Comment 4: The experiments should be performed in orthogonal methods to substantiate the conclusions. Immunoblots should be provided for Fig 1B, 2J, 5B and 5D. Please show p62 in immunoblot as well.

Response: Thanks for the comments. Additional western blot experiments were performed. The revised manuscript now contains new western blot results of LC3B, Bcl-2 subfamily proteins, Ki67

and p62, which confirm our findings that were made based on immunofluorescence. The new data has been incorporated in the revised manuscript, supplementary Figure S9 and S15.

Comment 5: Figure 4C: As presented; the NPs accumulate in near CD31+ tumor blood vessels. How does the drug reach in the CD31 negative areas? Do those areas maintain proliferation? A co-staining with Ki67 would be more informative.

Response: Thanks for the comment. We want to answer this question by making the following points.

- The experiment in Figure 4C was performed after single IV injection. In the efficacy studies, multiple IV injections were performed. We have provided low-magnification Ki67 IHC image (Figure S13) to show relatively homogenous anti-proliferation effect throughout the tumor, presumably due to particles' efficient intratumoral distribution post repeated IV injections.

- If a heterogeneous particle distribution becomes a major concern in other cancer models (or patients), we will contemplate the use of “two-wave” therapy to improve nanoparticle intra-PDAC distribution. In fact, we have developed an adjunct “first-wave” nanocarrier that decreases pericyte coverage of the PDAC vasculature through interference in the pericyte recruiting TGF- β signaling pathway. Sequential use of TGF- β inhibition nanocarrier, followed by drug delivery nanocarrier led to more abundant and homogenous nanocarrier distribution and improved efficacy in a stroma-rich human PDAC model in mice (ACS Nano 2013, 7, 11, 10048). A brief discussion was added in the end of the manuscript (page 15).

Comment 6: The histology sections are presented only in higher magnification. As this can be subjective, the authors should provide lower magnification views for a better assessment. A

proper quantification is missing and should accompany every histological analysis.

Response: Thanks for the comment. Low-mag pictures now appear as online data supplementary Figure S13 (Ki67 and pRB), S14 (CC-3), S16 (Bcl proteins) and S19 (Ki67 and pRB). We also added semi-quantification results in the revised manuscript.

Comment 7: Many studies have shown CDK4/6 inhibition alone or combined with other drugs invoke a senescence response. Senescence is associated with the upregulation of Bcl-2 family of proteins. In this study, the authors found Bcl-x_L is upregulated; however, senescence is not associated with their drug treatment. This is an important observation that needs to be substantiated with proper controls. A) In Fig S10, a proper positive control for SA-β-gal staining should be presented to show that the assay is working in their setting. B) As discussed before this phenomenon should be proved in multiple cell lines. C) The authors can perform quantitative RT-PCR to show that senescence-associated secretory phenotype (SASP) related transcripts are not regulated in their assays.

Response: As per reviewer's suggestion, the study of senescence has been solidified from the following aspects:

(i) Controls: Control groups, such as HCQ NP and PAL NP treatments, have been incorporated in the SA-β-gal assay (now Supplementary Figure S17). PAL NP induced distinct SA-β-gal expression, which functioned as positive control in this study. Contrast to PAL NP, co-delivery NP failed to induce SA-β-gal production in all 5 PDAC cell lines.

(ii) Additional cell lines: In addition to PANC-1 cells, SA-β-gal has been evaluated in multiple PDAC cells lines. The results in Supplementary Figure S17A suggested that co-delivery NP did not induce the production of SA-β-gal across a series of PDAC cells, i.e. MIA PaCa-2, BxPC-3, HPAF-II and AsPC-1 cells. The new results confirmed our PANC-1 data.

(iii) PCR analysis on senescence-associated secretory phenotype (SASP): Thanks for pointing this out. Compared to other cell stress, SASP is a "hot" area that is under robust investigation. While SA-β-gal assay is generally believed as a gold-standard for the identification of senescence *per se*, SASP pattern could add additional information, especially from the immunology aspect (Science 362, 6421, 1416). Literature has suggested that SASP is a dynamic process and is frequently cell type-, cancer type-, and stress-dependent. Microarray analysis of SASP suggested at least ~60 factors were influenced, including interleukins, chemokines, growth factors, secreted proteases, and ECM components, etc (Annu. Rev. Pathol. Mech. Dis. 5:99-118). Among the family of SASP, the most prominent cytokine of the SASP is interleukin-6 (IL-6), a pleiotropic pro-inflammatory cytokine (Annu. Rev. Pathol. Mech. Dis. 5:99-118). In PANC-1 pancreatic cells, SASP molecules (induced by gemcitabine) were reported, such as PML and DCR2 (Biochem Biophys Res Commun. 2016, 477, 515). Given this background, RT-PCR detection of IL-6, PML and DCR2 has been performed. While PAL-only particle indeed induces an increased level of SASP factors, the PCR results also suggest that co-delivery NP could not induce senescence in this particular case. This discovery is in agreement with the SA-β-gal staining results.

Collectively, different from PAL-only NP, HCQ/PAL co-delivery formulation did not trigger senescence, evidenced by SA-β-gal and SASP analyses. While this observation is interesting, the underlining mechanism is out of the scope of current work. We speculate that the pleiotropic effects of HCQ might play a role.

Comment 8: As discussed, PDAC has a unique fibrous micro-environment. Not only it is impermeable to drug formulations, but also the cross-talk of immune cell and tumor cell provide a challenge to treat PDAC. As authors claim a more potent drug delivery method, the efficacy of NP or NP+ABT737 should be performed in immunocompetent PDAC models (i.e. Kras-p53 GEMMs) or syngeneic orthotopic transplantation models.

Response: We have addressed this question above. In short, we would like to reserve the immunological study in the future. Based on our intensive experiences in studying nano-enabled chemo-immunotherapy in cancer, immunology experiments require separate planning and cannot be an “add-on” into this already comprehensive investigation. Moreover, we prefer to use a Rb⁺ model in immunocompetent mice as the primary model, rather than “Kras-p53 GEMMs” model (*a.k.a.* KPC model), which does not express Rb/pRb (see response in 2(d)). To accommodate the reviewer’s comment, we have added a section about the immunological effect of CDK4/6 inhibitor in the end of our manuscript.

We greatly appreciate the review and hope that we have fully addressed all concerns and critique.

Sincerely,

Huan Meng, PhD
Associate Professor
University of California, Los Angeles (UCLA)
Dept. of Medicine – Div. of NanoMedicine
Tel: 1-310-825-0217 (office) | 1-310-983-3359 (lab)
Email: hmeng@mednet.ucla.edu

REVIEWERS' COMMENTS:

Reviewer #1 (Remarks to the Author):

The authors have properly answered my concerns. I have no more questions for this manuscript.

Reviewer #2 (Remarks to the Author):

The authors have addressed the majority of the concerns. Now the paper is substantially improved with key details. With a recent surge of interest to use CDK4/6 inhibitor for pancreatic cancer treatment, this study offers an improved drug delivery method and a combinatorial drug formulation that may benefit human patients. The following minor concerns should be addressed in the revised manuscript.

1) In the rebuttal letter (comment 2D), the authors have shown that the KPC PDA cell line has low RB and phospho-RB(pRB) compared to the human cancer cell lines they have used in their study. They have expressed their skepticism on using KPC PDA animal models to investigate CDK4/6 inhibitors. However, employing KPC animal model, Ruscetti et al (Cell, Volume 181, Issue 2, 16 April 2020, Pages 424-441.e21) have shown that CDK 4/6 inhibitor (in combination with trametinib) can increase senescence through RB, and generate a sensitive background for immunotherapy. Hence it is not clear if the low RB/pRB showed by authors is an in-vitro phenomenon. The author should publish the low RB/pRB level in KPC-PDA cells in the supplemental Fig 1A with other RB Positive cell lines. As there is an emerging interest to test CDK4/6 inhibitors with various drug combinations for pancreatic cancer, this information can be useful to choose the right cell lines and help the scientific community in future research.

2)Comment 5: Figure 4C: For clarity, the authors should mention that the experiment B and C were done with single IV injection (or add “the experiment was performed as 4B” to the 4C figure legend).

Response Letter to Reviewers' Comments

We appreciate the reviewers' addition comments on our manuscript. Please see the point-to-point responses below.

Reviewer #1

Comment: The authors have properly answered my concerns. I have no more questions for this manuscript.

Response: Thank you. We are glad that we have addressed the concerns from Reviewer 1.

Reviewer #2

Comment: The authors have addressed the majority of the concerns. Now the paper is substantially improved with key details. With a recent surge of interest to use CDK4/6 inhibitor for pancreatic cancer treatment, this study offers an improved drug delivery method and a combinatorial drug formulation that may benefit human patients. The following minor concerns should be addressed in the revised manuscript.

Response: Thank you. We appreciate the reviewer's positive comments, i.e. "Now the paper is substantially improved with key details" and "the authors have addressed the majority of the concerns". The remaining minor concerns have been addressed below. Necessary revision was made accordingly.

Comment: 1) In the rebuttal letter (comment 2D), the authors have shown that the KPC PDA cell line has low RB and phospho-RB(pRB) compared to the human cancer cell lines they have used in their study. They have expressed their skepticism on using KPC PDA animal models to investigate CDK4/6 inhibitors. However, employing KPC animal model, Ruscetti et al (Cell, Volume 181, Issue 2, 16 April 2020, Pages 424-441.e21) have shown that CDK 4/6 inhibitor (in combination with trametinib) can increase senescence through RB, and generate a sensitive background for immunotherapy. Hence it is not clear if the low RB/pRB showed by authors is an in-vitro phenomenon. The author should publish the low RB/pRB level in KPC-PDA cells in the supplemental Fig 1A with other RB Positive cell lines. As there is an emerging interest to test CDK4/6 inhibitors with various drug combinations for pancreatic cancer, this information can be useful to choose the right cell lines and help the scientific community in future research.

Response: Thank you. Per reviewer's suggestion, Supplemental Figure S1 now includes multiple human PDAC cell lines plus murine KPC cells. We also added the following information in the figure caption of Fig. S1. It now reads: "... (B): For screening purpose,

a list of human PDAC cell lines and a KPC cell line (derived from a spontaneous PDAC tumor from a transgenic *Kras*^{LSL-G12D/+}*Trp53*^{LSL-R172H/+}*Pdx1-Cre* mouse) were assayed for Rb and pRb expression. All the human PDAC cell lines are positive for both Rb and pRb. KPC cells exhibit an extremely low Rb/pRb expression, which is similar to the literature (Gut. 2018, 267, 2142). Five Rb/pRb⁺ PDAC cell types, namely PANC-1, MIA PaCa-2, BxPC-3, HPAF-II and ASPC-1, are included in our investigation.”

Comment: Figure 4C: For clarity, the authors should mention that the experiment B and C were done with single IV injection (or add “the experiment was performed as 4B” to the 4C figure legend).

Response: Thank you. We have clarified that a single IV injection was performed in the biodistribution experiment in the Figs. 4B and 4C.

We greatly appreciate the review and hope that we have fully addressed all concerns and critique.

Sincerely,

Huan Meng, PhD

Associate Professor

University of California, Los Angeles (UCLA)

Dept. of Medicine – Div. of NanoMedicine

Tel: 1-310-825-0217 (office) | 1-310-983-3359 (lab)

Email: hmeng@mednet.ucla.edu